# Bacterial infection promotes tumorigenesis of colorectal cancer via regulating CDC42 acetylation

Dan-Ni Wang[1,☯], Jin-Jing Ni[1,☯], Jian-Hui Li[1,2], Ya-Qi Gao[3], Fang-Jing Ni[1], Zhen-Zhen Zhang[4], Jing-Yuan Fang[3], Jie Lu[5]*, Yu-Feng Yao[1,5,6,7]*

1 Laboratory of Bacterial Pathogenesis, Shanghai Institute of Immunology, Shanghai Jiao Tong University School of Medicine, Shanghai, China, 2 Shanghai Institute of Phage, Shanghai Public Health Clinical Center, Fudan University, Shanghai, China, 3 State Key Laboratory for Oncogenes and Related Genes; Division of Gastroenterology and Hepatology, Renji Hospital, Shanghai Jiao Tong University School of Medicine, Shanghai, China, 4 Department of Pathology, Renji Hospital, Shanghai Jiao Tong University School of Medicine, Shanghai, China, 5 Department of Infectious Diseases, Ruijin Hospital, Shanghai Jiao Tong University School of Medicine, Shanghai, China, 6 State Key Laboratory of Microbial Metabolism, and School of Life Sciences and Biotechnology, Shanghai Jiao Tong University, Shanghai, China, 7 Shanghai Key Laboratory of Emergency Prevention, Diagnosis and Treatment of Respiratory Infectious Diseases, Shanghai, China

☯ These authors contributed equally to this work.
* lj11750@rjh.com.cn (JL); yfyao@sjtu.edu.cn (Y-FY)

**Data Availability Statement:** All data generated or analyzed in this study are included in the manuscript and Supplementary Information.

## Abstract

Increasing evidence highlights the role of bacteria in promoting tumorigenesis. The underlying mechanisms may be diverse and remain poorly understood. Here, we report that *Salmonella* infection leads to extensive de/acetylation changes in host cell proteins. The acetylation of mammalian cell division cycle 42 (CDC42), a member of the Rho family of GTPases involved in many crucial signaling pathways in cancer cells, is drastically reduced after bacterial infection. CDC42 is deacetylated by SIRT2 and acetylated by p300/CBP. Non-acetylated CDC42 at lysine 153 shows an impaired binding of its downstream effector PAK4 and an attenuated phosphorylation of p38 and JNK, consequently reduces cell apoptosis. The reduction in K153 acetylation also enhances the migration and invasion ability of colon cancer cells. The low level of K153 acetylation in patients with colorectal cancer (CRC) predicts a poor prognosis. Taken together, our findings suggest a new mechanism of bacterial infection-induced promotion of colorectal tumorigenesis by modulation of the CDC42-PAK axis through manipulation of CDC42 acetylation.

## Author summary

Protein acetylation plays an important role in regulating various aspects of cell life. In this study, we show that *Salmonella* infection disturbs acetylation of more than 90 proteins, including Rho GTPase cell division cycle 42 (CDC42). CDC42 is involved in many crucial signaling pathways in cancer cells. We find that CDC42 K153 can be deacetylated by the NAD$^+$-dependent deacetylase SIRT2 after *Salmonella* infection, which causes an impaired

**Funding:** This work was supported by grants from the National Natural Science Foundation of China 31900111 (to DW), 81830068 (to YY), 81772140 (to YY), 81501733 (to JLu), and 31700121 to (to JN), Key Research and Development Project of China (No. 2016YFA0500600) (to YY), the Program for Professor of Special Appointment (Eastern Scholar) at Shanghai Institutions of Higher Learning (to YY), GuangCi Professorship Program of Ruijin Hospital, Shanghai Jiao Tong University School of Medicine (to YY). The funders had no role in study design, data collection and analysis, decision to publish, or preparation of the manuscript.

**Competing interests:** The authors have declared that no competing interests exist.

binding of its downstream effector PAK4. Low acetylation level of CDC42 K153 is crucial for tumor cell migration, invasion and apoptosis. Furthermore, lower K153 acetylation in tumor tissues compared to adjacent normal tissues of colorectal cancer (CRC) patients is correlated to the poor prognosis of CRC. Collectively, the information derived from this study suggests that bacterial infection could promote CRC tumorigenesis by modulating CDC42 acetylation.

## Introduction

*Salmonella* is one of the most common foodborne pathogens in the world [1], which can cause a variety of diseases in different hosts, from mild enterocolitis to systemic fatal infection [2–4]. Recent research indicates that there are more bacteria in tumor tissues than in normal tissues, and *Salmonella* has been found in tumor tissues of colorectal cancer (CRC), breast cancer, ovarian cancer, and even lung cancer [5]. CRC is considered the third most common cancer worldwide and the second leading cause of cancer-related deaths [6]. A comprehensive study showed that the content of bacteria in tumor tissues was significantly higher than that in adjacent normal tissues [5]. Moreover, bacteria in tumor tissues may promote cancer development [7]. For example, *Fusobacterium nucleatum* is a well-known bacterium that promotes the development of CRC through TLR4 /Kap1/NRF2 signal pathway [8]. Interestingly, several studies have shown that *Salmonella*, which causes digestive tract infection as the common symptom, may be related to the development of CRC. It has been reported that antibody levels against *Salmonella* flagellin were higher in CRC and pre-cancer cases than in controls, suggesting a possible link of *Salmonella* to CRC development [9]. The increased risk of CRC was observed among patients with *Salmonella* infection [10,11]. In addition, *Salmonella* effector protein AvrA could increase the incidence of colorectal tumors in a mouse CRC model [12].

Cell division cycle 42 protein (CDC42) is highly expressed in 60% of CRCs and plays an important role in cancer development [13,14]. CDC42 is a member of the Rho family of GTPases and is involved in the regulation of key cell functions [15]. CDC42 plays a critical role in several cell signaling pathways, and therefore, many pathogens hijack CDC42 to facilitate their infection [16]. A previous study showed that *Salmonella* infection can induce host cell actin rearrangement to promote bacterial internalization by activating CDC42 [17]. The activated CDC42 can regulate the expression of the family of p21-activated protein kinases (PAKs), which are generally highly expressed in tumor cells [18]. PAKs can activate several signaling pathways, including the mitogen-activated protein kinase (MAPK) signaling pathway, thus promoting the development of tumors [19,20]. According to the biochemical and structural features, PAKs are classified into two groups: group I and group II [21]. Generally, group I PAKs (PAK1-3) are in a self-inhibiting state under physiological conditions and are activated by binding to CDC42, and cancer development could be promoted by the CDC42-PAK1 axis [22]. Group II PAKs (PAK4-6) preferentially bind to CDC42 under physiological conditions [23]. Because group II PAKs do not have a self-inhibition domain, the activated group II PAKs do not need to combine with CDC42 [21]. However, recent studies have demonstrated that group II PAKs can be activated by CDC42 [24,25]. Interestingly, *S.* Typhimurium infection can block the binding of CDC42 and PAK4 [26]. The activated PAKs promote tumor development by driving important signaling pathways, including the MAPK and phosphoinositide 3-kinase (PI3K/AKT) pathways [20,25].

Many proteins in colorectal tumor tissues show abnormal post-translational modifications (PTMs) [27,28], and dysregulation of PTMs has been implicated in various cancers [29]. As a

major type of PTMs, acetylation not only plays an important role in promoting CRCs but also functions as a molecular marker for cancer diagnosis [27,30,31]. Acetylation is also vital for both bacterial virulence and multiple host cell processes [32,33].

As an enteropathogenic bacterium, *S*. Typhimurium affects various cell biological processes by regulating PTMs of host proteins [34–37]. For example, the *Salmonella* effector AvrA inhibits host inflammatory responses by acetylating p53 [38]. To understand the role of acetylation induced by *S*. Typhimurium infection in host cells, we infected human U937 and THP-1 monocytes with *S*. Typhimurium and identified the acetylated proteins with stable isotope labeling by amino acids in cell culture (SILAC). The results showed that *S*. Typhimurium infection led to acetylation change in an array of proteins, including CDC42. We systematically investigated the role of CDC42 acetylation in CRC and revealed how *Salmonella* infection regulated CDC42 activity and its downstream signaling pathways through acetylation. The findings of this study shed new light on bacterial involvement in CRC progression.

## Results

### *Salmonella* infection lowers CDC42 K153 acetylation

Pathogens can hijack the host signaling pathways by regulating host protein acetylation [34]. To identify the host protein lysine acetylome in response to *Salmonella* infection, we performed LC/MS-based quantitative analysis of lysine acetylome. This approach combined SILAC with anti-acetyl-lysine (anti-KAc) antibody enrichment of peptides after tryptic digestion of proteins, and the workflow for identification of KAc peptides from cell lysates is shown in Fig 1A. Briefly, heavy isotope-labeled human U937 and THP-1 cells were infected with *S*. Typhimurium, and light isotope-labeled mock-infected human U937 and THP-1 cells were used as control. To determine *Salmonella* infection-induced dynamic changes in the host protein acetylome, the cells were harvested at 1, 3, and 24 h post-infection. The extracted heavy and light isotope-labeled proteins were combined in a ratio of 1:1, and tryptic peptides were subjected to sequential anti-KAc enrichment and 1D-Q-Exactive LC/MS analysis with two replicates (S1 Table). We summed up eXtracted Ion Current (XIC) of all isotopic clusters associated with the identified amino acids sequence in the light label cluster and heavy partner and calculated the ratio between two heavy and light label partners. We also assessed the significance of outlier ratios using Significance A. Only peptides with average fold change of acetylation between two states greater than 1.5 and Significance A values less than 0.05 were considered up or down regulated significantly by *Salmonella* infection. Finally, a total of 814 KAc sites in U937 and THP-1 cells were determined quantitatively. From the six repetitive experiments, we identified 62 KAc sites from 40 proteins with Heavy: Light (H:L) ratios $\geq$ 1.5 and 72 KAc sites from 57 proteins with H:L ratios $\leq$ 0.7 (*i.e.* L:H ratios $\geq$ 1.5) (S1A Fig). Functional annotation analysis of these dynamic KAc sites revealed enrichment of cellular metabolism, cellular response to stress and posttranscriptional regulation of gene expression (Fig 1B).

Interestingly, our SILAC results revealed that several members of the Rho GTPases network (CDC42, FAS, and MYC) underwent dynamic acetylation following *Salmonella* infection (Fig 1C). Some pathogenic bacteria, including *Salmonella*, *Yersinia* and *E. coli*, produce an array of virulence factors that target Rho proteins. These pathogens exploit and/or impair many aspects of Rho protein activities by activating or inhibiting these key molecular switches [39]. We then focused on CDC42 to study the role of acetylation of CDC42 in *Salmonella* infection. The SILAC results showed that CDC42 acetylation at lysine 153 (K153) was downregulated by approximately 2-fold at 1 h post-infection (Fig 1D). To confirm this observation, we generated a site-specific antibody to K153-acetylated (K153Ac) CDC42, and the specificity of this antibody was verified by a dot-blot assay (S1B Fig). Flag-tagged CDC42 was then

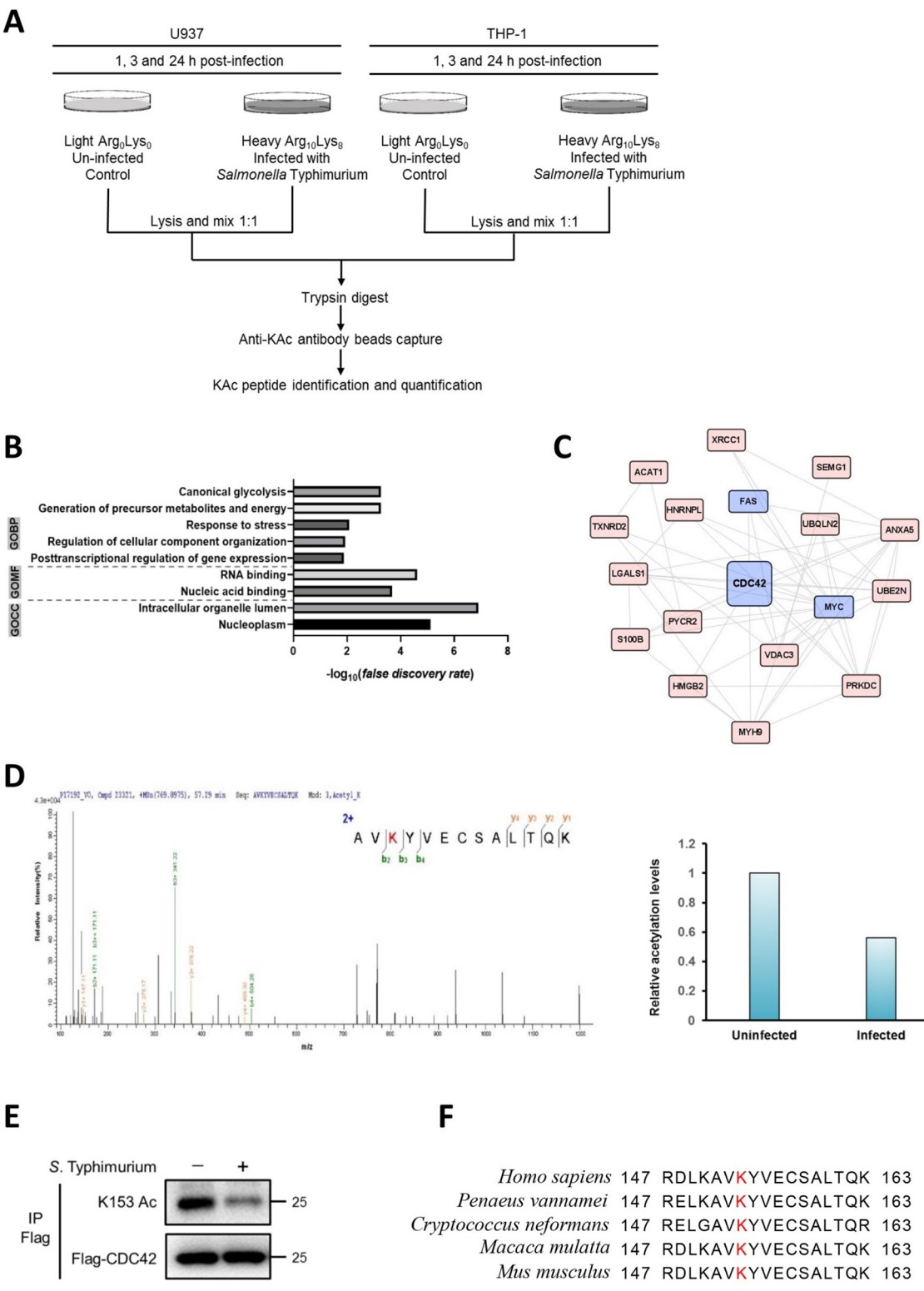

**Fig 1. *Salmonella* infection changes host protein acetylation.** (A) Lysine acetylomics workflow is shown. Cell pellets were lysed, combined, reduced, alkylated, and digested with trypsin to generate peptides with 5 mg of proteins input per state. Then, digested peptides were subjected to anti-KAc antibody enrichment and LC/MS analysis. (B) Annotation enrichment analysis of proteins with regulated KAc sites identified in U937 and TPH-1 cells infected with *S.* Typhimurium for 1, 3, and 24 h. The bar graphs show significantly gene ontology biological processes (GOBPs), gene ontology cellular compartments (GOCCs) and gene ontology

molecular functions (GOMFs). (C) Associations among proteins with regulated Kac sites based on STRING. (D) Representative mass spectrometry analysis of the peptide containing CDC42 K153Ac. Shown is the spectrum covering the region from 100 to 1200 m/z which includes the peptide containing the acetylated lysine 153. Acetylation of CDC42 K153 was downregulated by ~2-fold at 1 h. (E) Verification of K153 acetylation level after *Salmonella* infection. Flag-CDC42 was transfected into HEK293T cells for 48 h, followed by *S.* Typhimurium infection for 1 h. Cell lysates were used for IP with anti-Flag antibody, and K153 acetylation was determined by western blot (WB) assay with CDC42-specific acetyl-K153 antibody (anti-K153Ac). (F) Alignment of CDC42 amino acid sequences from various species, red box indicates the conserved K153.

overexpressed in HEK293T cells with or without *S.* Typhimurium infection. Western blot assay showed that the acetylation of CDC42 K153 in *Salmonella*-infected cells was significantly lower than that in uninfected cells (Fig 1E). We also found that K153 of CDC42 was highly conserved in various species (Fig 1F), which suggests that CDC42 K153 acetylation may be universally present.

To investigate whether infection of other bacterial species can manipulate CDC42 K153 acetylation, we infected HEK293T cells overexpressing Flag-CDC42 with several enteropathogenic and probiotic bacteria. The result showed that the acetylation levels of CDC42 K153 decreased in cells infected with enteropathogenic bacteria, including *S.* Typhimurium, *F. nucleatum*, *Enterococcus faecalis*, Enteropathogenic *Escherichia coli* (EPEC), and *Listeria monocytogenes* (S1C Fig). *F. nucleatum*, *E. faecalis*, and EPEC have been reported to promote the occurrence and development of CRC [7,40]. In contrast, most probiotics did not affect the acetylation of CDC42 K153, such as *Streptococcus thermophiles*, *Lactobacillus rhamnosus*, *Lactobacillus plantarum*, and *Lactobacillus paracasei* (S1C Fig), and these bacterial species have been shown to inhibit tumor growth of CRC [41–44].

## Acetylation of CDC42 K153 promotes its binding to PAK4

A previous study showed that the interaction between CDC42 and PAK4 is attenuated after *S.* Typhimurium infection [26]. We analyzed the crystal structure of PAK4 in complex with CDC42 (PDB-ID: 5UPK) and found that CDC42 E171 and PAK4 R489 residues may interact with each other [45], and the side chain of CDC42 residue K153 is close to that of E171 (3.9 Å) (Fig 2A). Because lysine acetylation leads to neutralization of its positive electrostatic charge, we speculate that deacetylation of CDC42 K153 may expose the positive charge on this lysine side chain, stabilize its interaction with CDC42 E171, and thus abolish the binding between CDC42 E171 and PAK4 R489. To test this hypothesis, HEK293T cells were co-transfected with CDC42 and PAK1 or PAK4 followed by *S.* Typhimurium infection. The protein interactions were determined by co-immunoprecipitation (co-IP) and immunoblotting assays. The results showed that *S.* Typhimurium infection drastically decreased the binding between CDC42 and PAK4, but barely affected the binding between CDC42 and PAK1 (Fig 2B).

Next, we tested the role of K153 acetylation in CDC42 binding to PAKs by point mutation. To avoid the interference of endogenous CDC42 in HEK293T cells, CDC42 was knocked down by shRNA (S2A Fig). Single point mutation of lysine to arginine (R) mimicking the non-acetylated form attenuated the interaction between CDC42 and PAK4, while lysine to glutamine (Q) mutation, mimicking the constitutively acetylated form, resulted in enhanced binding of CDC42 to PAK4 (Fig 2C). Neither Q nor R mutation of K153 affected the binding of CDC42 to PAK1 (Fig 2D). Consistent with the findings of a previous study [46], *S.* Typhimurium infection enhanced the activity of CDC42 (S2B Fig), but no differences were found among GTP-bound CDC42 WT, K153Q, and K153R (Fig 2E). These results indicated that K153 de/acetylation affected the binding of CDC42 to PAK4, but did not affect the activation of CDC42 itself during *Salmonella* infection.

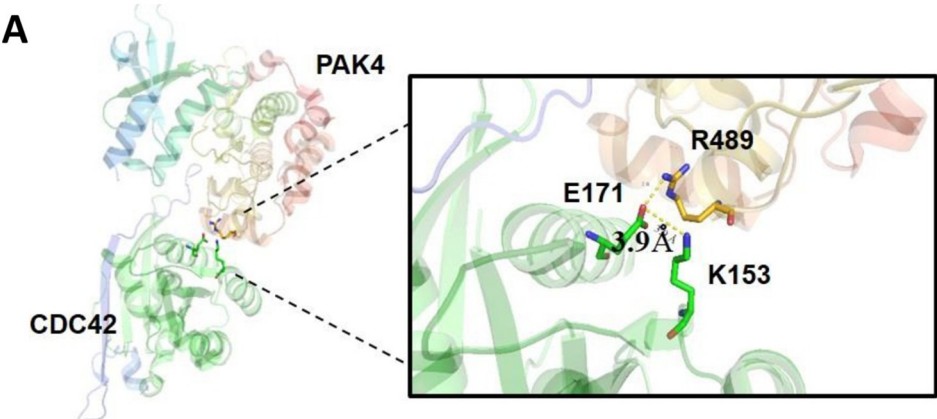

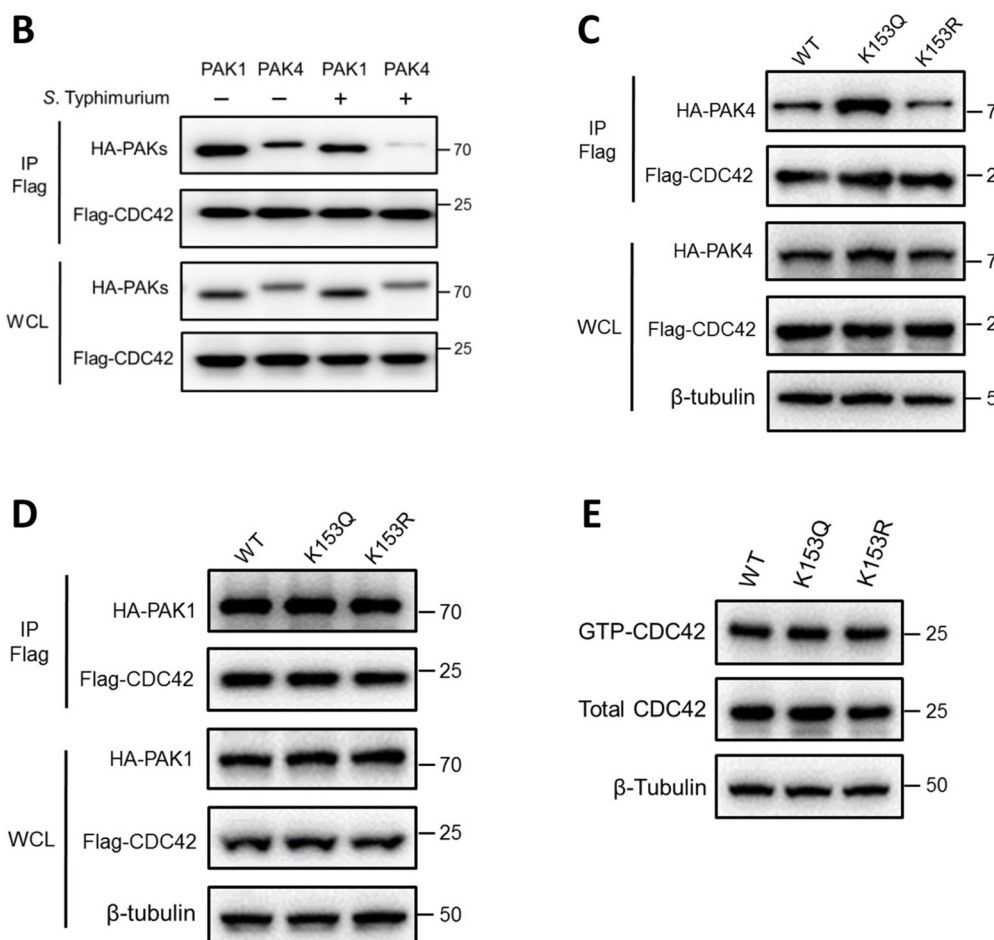

**Fig 2. Acetylation of K153 is essential for CDC42 interaction with PAK4.** (A) Ribbon diagram showing the structure of CDC42-PAK4 (PDB-ID: 5UPK). (B) *S.* Typhimurium infection drastically decreased the binding between CDC42 and PAK4. Flag-CDC42 was co-transfected with HA-PAK4 or HA-PAK1 into HEK293T cells for 48 h, followed by *S.* Typhimurium infection for 1 h. Protein interactions was analyzed by immunoprecipitation (IP) with anti-Flag antibody and followed by WB. K153R mutant attenuated the interaction between CDC42 and PAK4. Flag-

tagged CDC42-WT, K153Q or K153R was co-transfected with HA-PAK4 (C) or HA-PAK1 (D) into HEK293T cells with CDC42 knockdown. The association between CDC42 and PAK4 or PAK1 was determined by IP/WB with indicated antibodies. (E) K153 acetylation did not affect the activation of CDC42. Expression level of the active GTP-bound form of CDC42 WT or K153QR mutants in HEK293T cells with endogenous CDC42 knockdown was detected by CDC42 Activation Assay Kit (CST).

### SIRT2 deacetylates CDC42 K153

To identify the deacetylase of CDC42 K153, HEK293T cells overexpressing Flag-CDC42 were treated with the histone deacetylase (HDAC) family I/ II/ IV inhibitor trichostatin (TSA) or the sirtuin (SIRT) family deacetylase inhibitor nicotinamide (NAM), and CDC42 K153 acetylation was determined by western blot assay. We found that the acetylation level of K153 in cells treated with NAM was higher than that in cells treated with TSA (Fig 3A), which suggests the deacetylase of CDC42 K153 belongs to the SIRT family.

The SIRT family has seven members, SIRT1 to SIRT7, which are nicotinamide-adenine dinucleotide (NAD$^+$)-dependent deacetylases. Among them, SIRT2 resides predominantly in the cytoplasm and is highly expressed during bacterial infection, thereby mediating host immune responses [47,48]. SIRT1 is predominantly localized in the nucleus, although it can be found in the cytoplasm. The other SIRT family deacetylases are mainly located in the nucleus or mitochondria [49,50]. CDC42 is typically localized in cellular membranes [51]; therefore, CDC42 is more likely to interact with deacetylase in the cytoplasm. We co-transfected Flag-tagged CDC42 with various deacetylases into HEK293T cells and determined the interaction of CDC42 and deacetylases by co-IP and western blot assays. SIRT2, instead of other deacetylases interacted with CDC42 (S3A–S3E Fig). The acetylation level of K153 was reduced by SIRT2 treatment (Fig 3B). These findings suggest that SIRT2 is a deacetylase of CDC42 K153.

We incubated bacterial-expressed glutathione S-transferase (GST)-tagged CDC42 with cell lysates from HEK293T cells overexpressing SIRT2 and observed a sharp decrease in K153 acetylation as compared to that in the control (Fig 3C). AK7 (a SIRT2 inhibitor) treatment restored K153 acetylation (Fig 3D). In HEK293T cells with stable knockdown of SIRT2 (S3F Fig), *Salmonella* infection failed to reduce K153 acetylation (Fig 3E) and the binding of CDC42 to PAK4 was restored (Fig 3F). Taken together, both *in vivo* and *in vitro* results support that SIRT2-catalyzed deacetylation of CDC42 K153 undermine the binding of CDC42 with PAK4.

### p300/CBP acetylates CDC42 K153

Under physiological conditions, CDC42 K153 is acetylated in many cells and tissues [52]. To identify the acetyltransferase of K153, HEK293T cells were individually transfected with HA-tagged PCAF, Tip60, ACAT1, Gcn5, CBP, or p300. The results of western blot assay showed that CDC42 K153 could be acetylated by CBP or p300 (Fig 4A and 4B), but not by other acetyltransferases (S4A–S4D Fig). In addition, the specific inhibition of p300/CBP by A-485 treatment decreased the acetylation level of K153 and attenuated the binding of CDC42 to PAK4 (Fig 4C). We purified bacterial expressed GST-CDC42 and incubated with cell lysates from HEK293T cells overexpressing p300 or CBP. We found that the acetylation level of K153 was greatly increased by incubation with lysates containing p300/CBP (Fig 4D). These findings indicate that the p300/CBP is responsible for acetylation of CDC42 K153.

### Non-acetylated CDC42 K153 suppresses phosphorylation of p38 and JNK

A previous study reported that CDC42 is highly expressed in 60% of human CRCs, and its expression level is positively correlated with poorly differentiated CRCs [14]. Tumor cells usually have strong capabilities of proliferation, migration, and invasion. The growth and

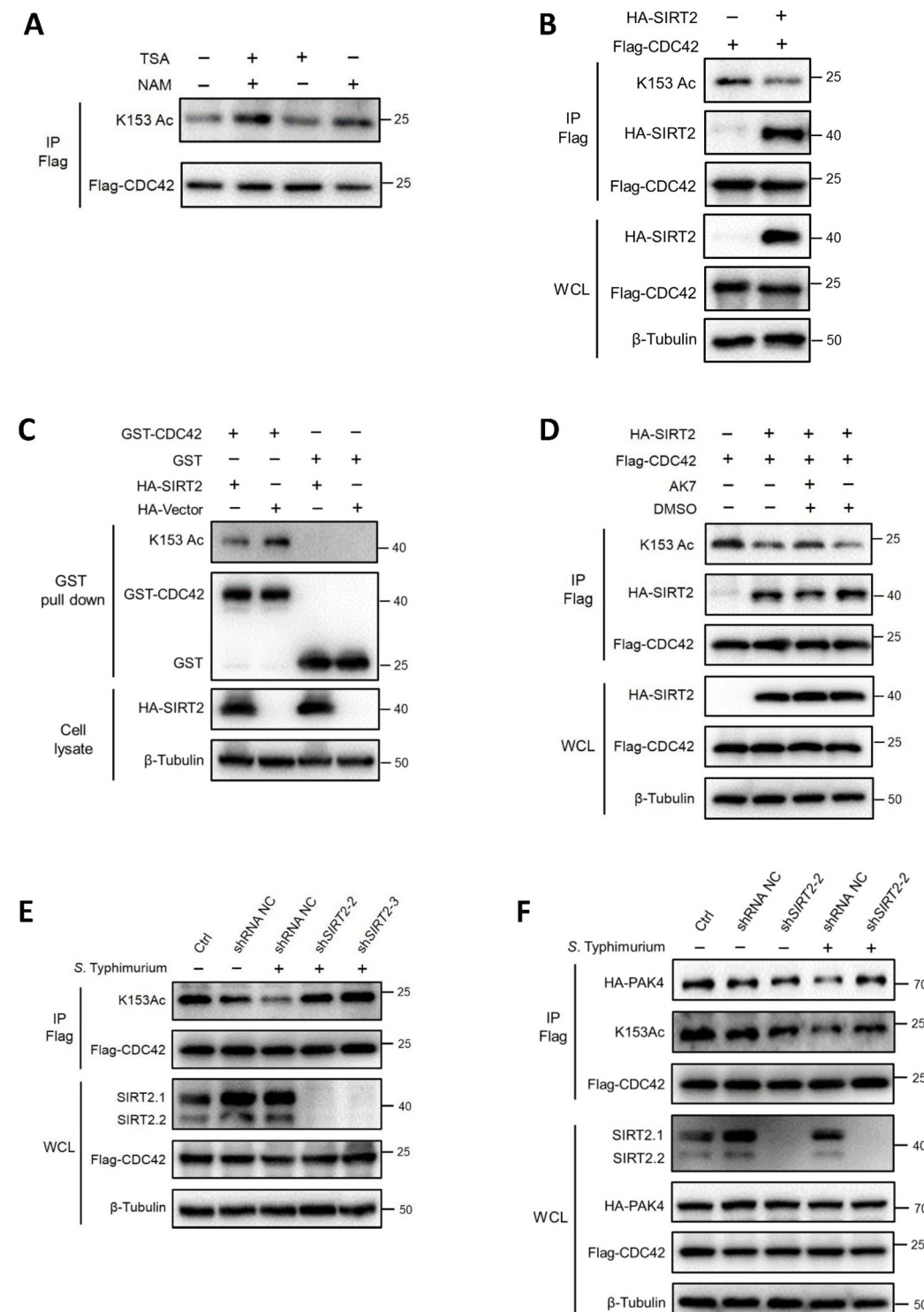

**Fig 3. CDC42 K153 is deacetylated by SIRT2.** (A) CDC42 K153 acetylation is regulated by SIRT family deacetylases. HEK293T cells transfected with Flag-CDC42 were treated with the deacetylase inhibitors TSA (2 μM) and NAM (10 mM) for 16 h before harvesting. K153 acetylation was determined by IP/WB. (B) SIRT2 decreases K153 acetylation *in vivo*. Overexpression of SIRT2 significantly decreases K153 acetylation. Flag-CDC42 was co-transfected with or without HA-SIRT2 into HEK293 cells, and K153 acetylation was analyzed by IP with anti-Flag antibody and WB with anti-K153Ac antibody. (C) SIRT2 decreases K153 acetylation *in vitro*. Purified GST-CDC42 protein was incubated with the cell lysate of HEK293T cells expressing HA-SIRT2 or the control vector. K153 acetylation was determined by the GST pull-

down assay, followed by WB with anti-K153Ac antibody. (D) A reduced acetylation of K153 was shown in AK7-treated cells. Flag-CDC42 was co-transfected with or without HA-SIRT2, followed by treatment with the SIRT2 specific inhibitor AK7 (10 μM) or dimethyl sulfoxide (DMSO) (as a negative control) for 16 h. K153 acetylation was then determined by IP/WB. SIRT2 knockdown increases K153 acetylation. Flag-CDC42 was transfected with or without HA-PAK4 into HEK293T cells with SIRT2 knocked down, and the cells were infected with *S*. Typhimurium. *Salmonella* infection failed to reduce K153 acetylation and binding between CDC42 with PAK4 in the SIRT2 knockdown cells. Cell lysates were used for IP with anti-Flag antibody. K153 acetylation was then analyzed by WB with anti-K153Ac antibody (E), and the association between CDC42 and PAK4 was determined by WB (F).

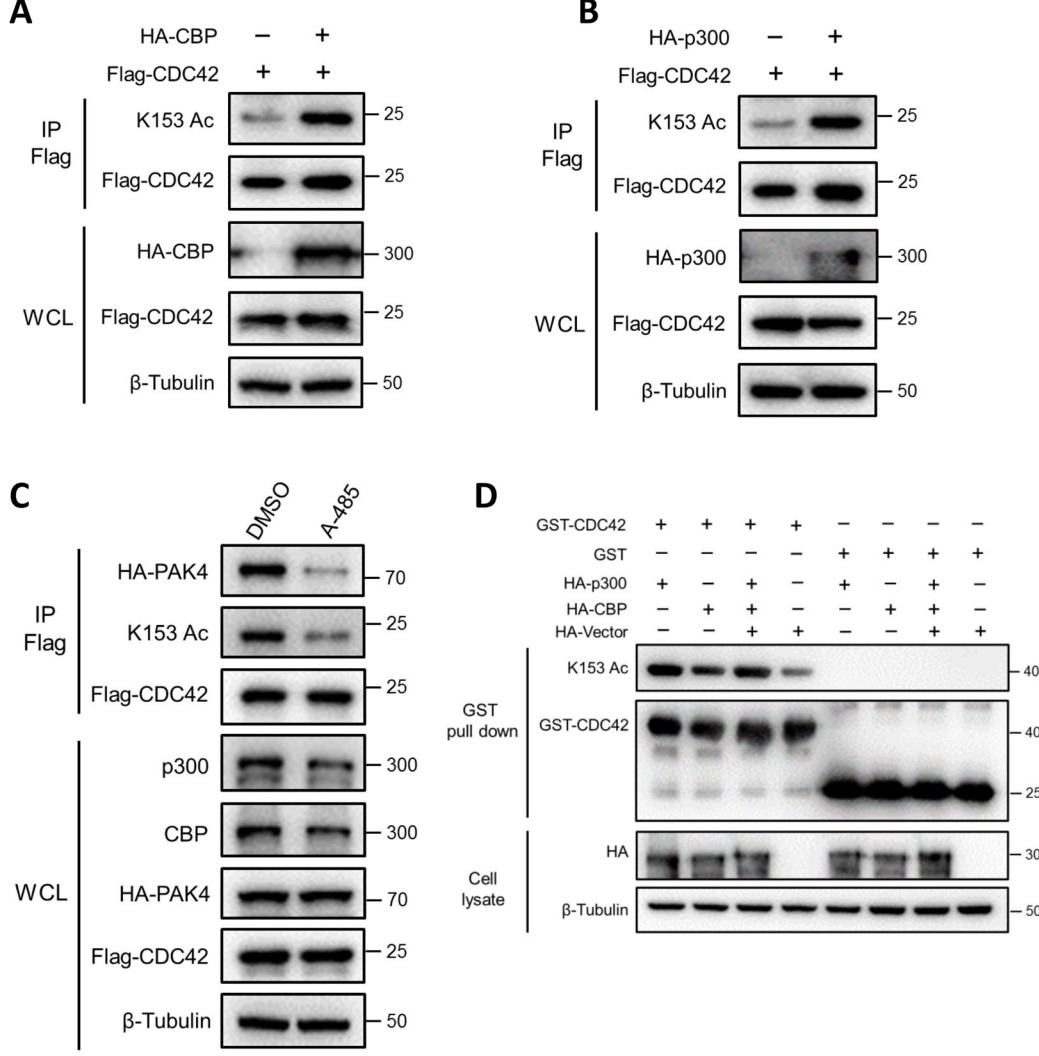

**Fig 4. CDC42 K153 is acetylated by p300/CBP.** CBP/p300 increases the acetylation level of K153. HA-CBP (A) or HA-p300 (B) was co-transfected with Flag-CDC42 into HEK293T cells, followed by IP with anti-Flag antibody and K153 acetylation level was examined by WB with anti-K153Ac antibody. (C) The p300/CBP-specific inhibitor A-485 attenuated K153 acetylation. Flag-CDC42 was co-transfected with HA-PAK4 into HEK293T cells, followed by treatment with A-485 (3 μM) or DMSO for 24 h before harvesting. The acetylation levels of K153 and the association between CDC42 and PAK4 were analyzed by IP/WB. (D) p300/CBP increases K153 acetylation *in vitro*. Purified GST-CDC42 protein was incubated with the cell lysate of HEK293T cells expressing HA-p300, HA-CBP, or the control vector. K153 acetylation was determined by GST pull-down and followed by WB.

differentiation of cells are closely associated with the PI3K/AKT/MAPK signaling pathway. To identify the role of CDC42 K153 acetylation in these signaling pathways, the endogenous CDC42 of HCT116 cells was knocked-down by shRNA (S5A Fig), and HCT116 cells stably expressing Flag-CDC42 WT, Flag-CDC42 K153Q or Flag-CDC42 K153R were established individually (Fig 5A). The results of western blot assay showed that the phosphorylation levels of JNK and p38 in cells expressing CDC42 K153R were lower than those in cells expressing CDC42 WT or CDC42 K153Q (Fig 5B), while the phosphorylation levels of ERK or AKT did not change in these K153 variant cells (Fig 5C). These results suggest that the phosphorylation level of JNK and p38 may be affected by the acetylation level of K153. To further reveal the effect of CDC42 K153 acetylation on the phosphorylation level of JNK and p38, CDC42 WT-, K153Q-, or K153R-overexpressing HCT116 cells were infected with *Salmonella*. The result showed that CDC42 WT-, K153Q-, and K153R-overexpressing cells had similar phosphorylation levels of JNK after *Salmonella* infection, but p38 still had a lower phosphorylation level in CDC42 K153R-overexpressing cells than in CDC42 WT- or CDC42 K153Q-overexpressing cells (Fig 5D). These results suggest that the acetylation of CDC42 K153 regulated the phosphorylation of p38 during *Salmonella* infection.

PAKs, as important downstream effectors of CDC42, are involved in regulating the phosphorylation level of JNK and p38 [53–55]. In the kinase domain, phosphorylation of the unique threonine residue (Thr423 in human PAK1 and Thr474 in human PAK4) is essential for full catalytic activity [56,57]. Therefore, we speculated that the effect of CDC42 K153R acetylation on the phosphorylation of JNK and p38 might be mediated by PAK. We then assessed the endogenous activity of PAKs in combination with different K153 lysine mutants of CDC42. The results showed that the amount of active PAK4 (p-PAK4(Thr474)) was less in K153R-overexpressing cells than in WT K153- or K153Q-overexpressing cells, but the amount of active PAK1 was comparable in these 3 cell lines (Fig 5E). This result suggests that the low level of K153 acetylation affected the binding of CDC42 with PAK4, thereby inhibiting the activation of PAK4. To further confirm that the relationship between PAK4 and the phosphorylation of JNK and p38, a PAK4 stable knockdown HCT116 cell line was constructed (S5B Fig). The results confirmed that the phosphorylation levels of JNK and p38 were reduced after PAK4 knock down (Fig 5F), and the phosphorylation level of p38 was still lower in PAK4 knockdown cells even after *Salmonella* infection (Fig 5G). This finding indicates that the acetylation level of K153 was positively correlated with the PAK4-dependent phosphorylation of JNK and p38, which may be related to the activity of PAK4.

## Non-acetylated CDC42 K153 increases cell survival, promotes tumor cell migration and invasion

Next, we sought to explore the tumorigenesis phenotype differences among K153 variants. Many studies have shown that JNK and p38 are involved in cell apoptosis [58–60]. The initiator caspases trigger a cascade-like proteolytic stimulation of effector caspase-3 and caspase-7 zymogens that subsequently drive the execution phase of the apoptotic cell death program by cleaving a plethora of functionally critical proteins within a cell, such as poly ADP-ribose polymerase (PARP) [61]. To reveal the role of CDC42 K153 acetylation in colorectal cell apoptosis, we used staurosporine (STS) to induce apoptosis of HCT116 cells expressing Flag-CDC42 WT, K153Q, or K153R. The results of western blot assay showed that the levels of cleaved caspase-3 and PARP in cells expressing CDC42 K153R were reduced as compared to those in the other two cell lines (Fig 6A). Phosphatidylserine (PS) is externalized to the plasma membrane as an "eat-me" signal for efferocytosis of apoptotic cells by phagocytes [61], hence, we assessed the role of K153 acetylation in cell surface PS exposure during STS-induced apoptosis. The

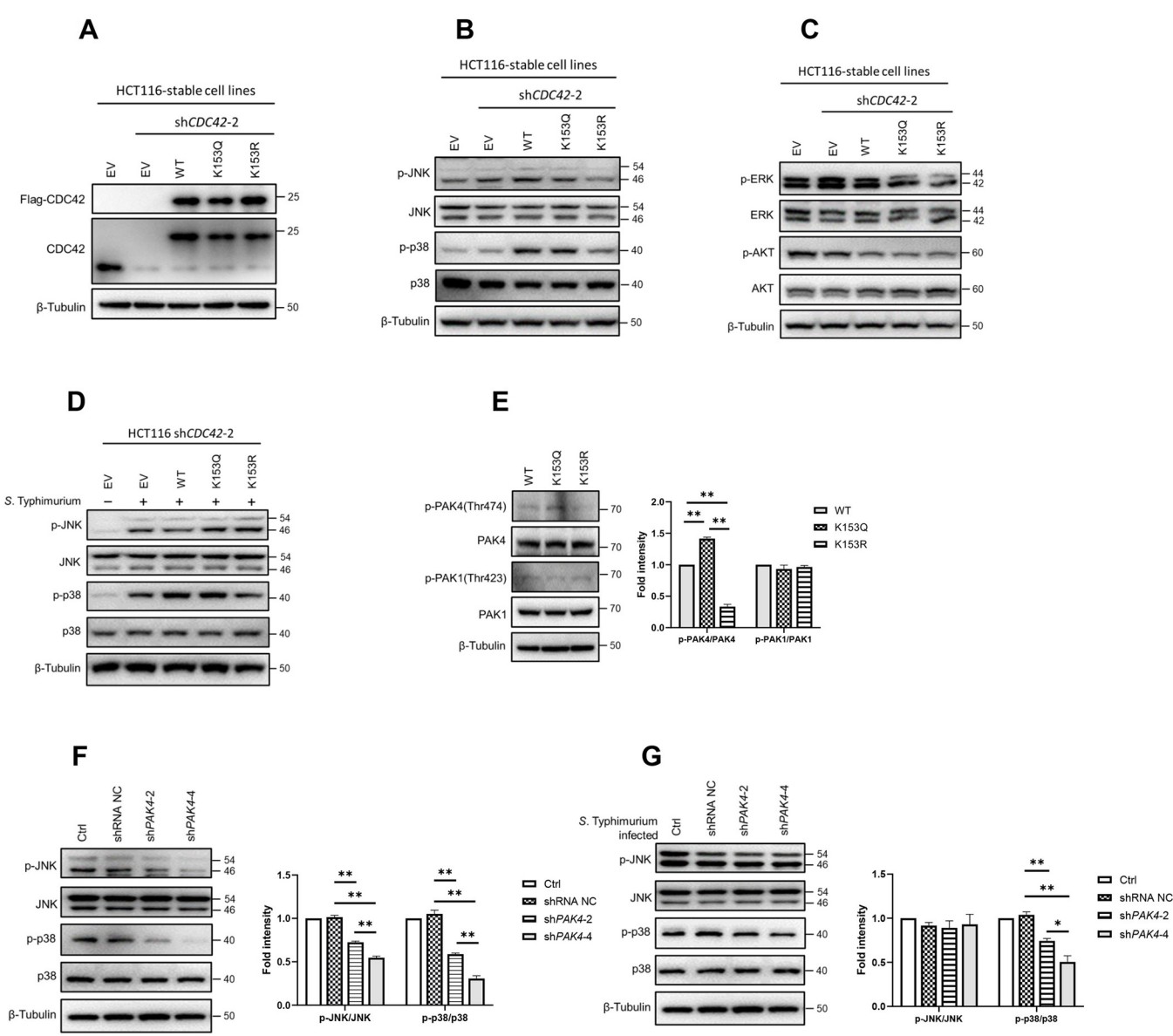

**Fig 5. Acetylation of CDC42 K153 affects JNK and p38 phosphorylation.** (A) HCT116 cells stably expressing Flag-CDC42-WT, Flag-CDC42 K153Q, or Flag-CDC42 were established. pCDH-flag vector (EV) was used as a control, and the cells were assessed by WB with anti-Flag antibody and anti-CDC42 antibodies. (B) Phosphorylation of JNK and p38 in K153R mutant. Phosphorylation of JNK and p38 were analyzed by WB in HCT116 cells stably expressing Flag-CDC42-WT, Flag-CDC42 K153Q, or Flag-CDC42 K153R, with pCDH-Flag vector (EV)-expressing cells as a control. (C) Determination of p44/42 (ERK) and AKT phosphorylation by WB in HCT116 cells stably expressing Flag-CDC42-WT, Flag-CDC42 K153Q, or Flag-CDC42 K153R, with pCDH-Flag vector (EV)-expressing cells as a control. (D) The phosphorylation of JNK and p38 were analyzed by WB in HCT116 cells stably expressing Flag-CDC42-WT, Flag-CDC42 K153Q, or Flag-CDC42 K153R treated by *S.* Typhimurium for 1 h. (E) Phosphorylation of PAK1 and PAK4 were determined in HCT116 cells stably expressing CDC42-WT, CDC42 K153Q or CDC42 K153R by WB (Left), and densitometry analysis was shown (Right). Phosphorylation levels of JNK and p38 were decreased in PAK4 knockdown cells, and a lower phosphorylation level of p38 in PAK4 knockdown cells after *Salmonella* infection. HCT116 cells with PAK4 knocked down were infected without (F) or with (G) *S.* Typhimurium for 1 h, and wild-type HCT116 cells was used as a control. JNK and p38 phosphorylation were analyzed by WB (Left), and densitometry analysis was shown (Right). $^*p<0.05$, $^{**}p<0.01$.

kinetics of PS externalization in HCT116 cells expressing CDC42 WT, CDC42 K153Q, or CDC42 K153R were determined by flow cytometry using Annexin V (AV) and propidium iodide (PI). The number of AV+PI- cells (21.96% ± 0.51%) in the K153R group was significantly less than those in the K153Q group, and the number of AV+ cells (30.49% ± 0.42%)

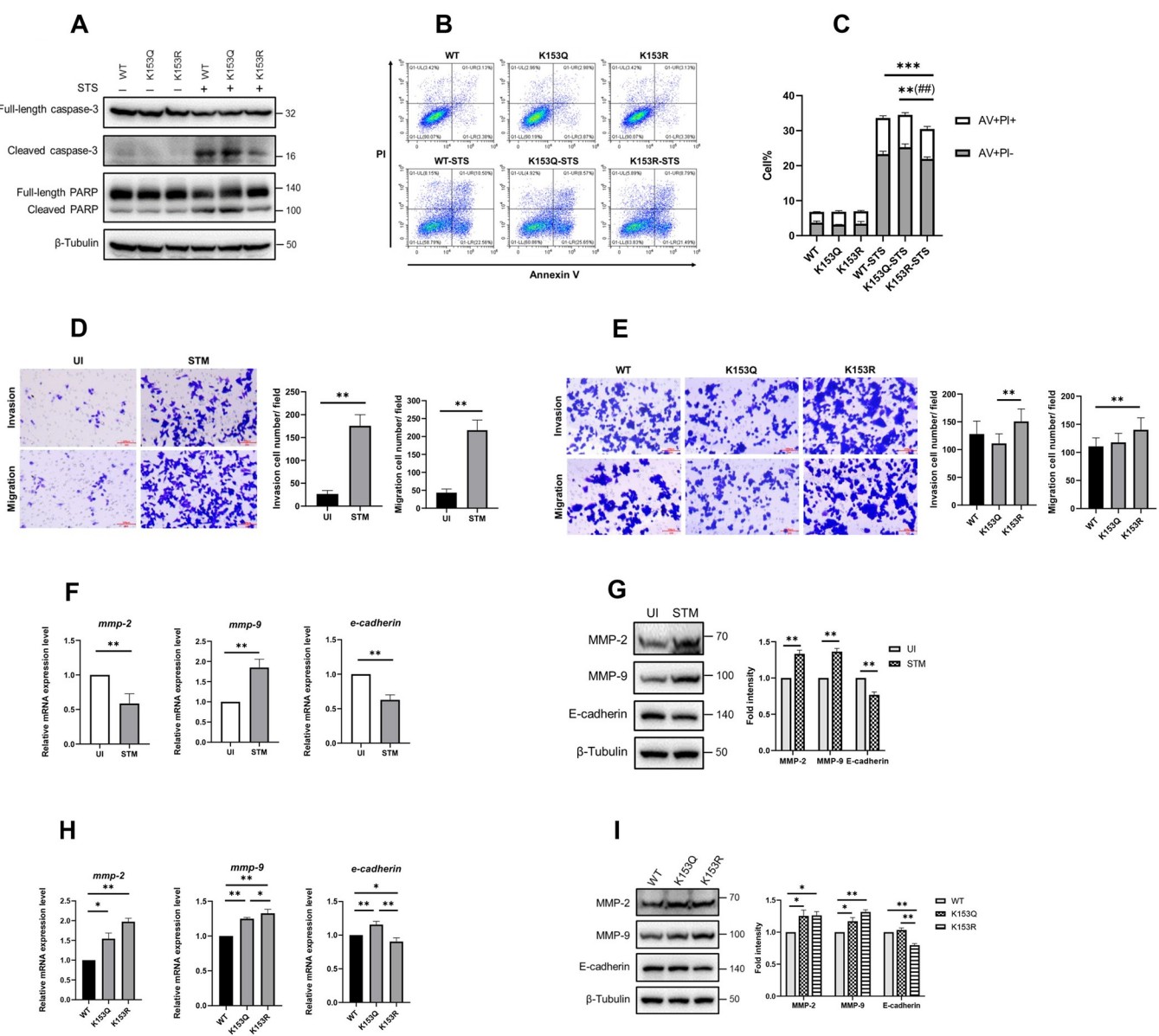

**Fig 6. Unacetylated CDC42 K153 is correlated with enhanced tumorigenesis phenotype of HCT116 cells.** HCT116 cells stably expressing CDC42-WT, CDC42 K153Q or CDC42 K153R were analyzed for apoptosis as followed: (A) The level of full-length or cleaved caspase-3 or PARP in cells treated with 4 μM staurosporine (STS) for 4 h was determined by WB. (B) Cells were treated with 0.4 μM STS for 24 h, and the level of surface exposure of the apoptosis signature phospholipid phosphatidyserine (PS) was determined by Annexin V (AV) staining followed by flow cytometry analysis. Propidium iodide (PI) was used to exclude necrotic cells. (C) Percentage of AV+PI- (gray bars) and AV+PI+ (white bars) cells. Asterisks represent *p* values for all AV+ cells, and octothorpes represent *p* values for AV+ and PI- cells (**$p<0.01$, ***$p<0.001$, ##$p<0.01$). (D) Migration and invasion assays of HCT116 cells infected with *S*. Typhimurium. (E) Migration and invasion assays of HCT116 cells stably expressed CDC42-WT, CDC42 K153Q, or CDC42 K153R. (F) Determination of the mRNA expression level of *mmp-2*, *mmp-9*, and *e-cadherin* by quantitative real-time PCR in HCT116 cells infected with *S*. Typhimurium. (G) Determination of MMP-2, MMP-9, and E-cadherin by WB in HCT116 cells during *S*. Typhimurium infection, and densitometry analysis was shown (Right). **$p<0.01$. (H) Determination of the mRNA expression level of *mmp-2*, *mmp-9*, and *e-cadherin* by quantitative real-time PCR in HCT116 cells stably expressing WT or QR mutants of CDC42 K153. (I) Determination of MMP-2, MMP-9, and E-cadherin by WB in HCT116 cells stably expressing CDC42-WT, CDC42 K153Q or CDC42 K153R, and densitometry analysis was shown (Right). *$p<0.05$, **$p<0.01$.

in the K153R group was significantly lower than those in the other two groups (Fig 6B and 6C). These results indicated that the low level of K153 acetylation may be beneficial for cell survival.

Previous studies have shown that bacterial pathogens could promote the development of CRC [62,63]. *S*. Typhimurium can manipulate the cell cycle of epithelial cells and macrophages [64]. Hence, we investigated the effect of *Salmonella* infection on cell cycle in HCT116 cells. The results of flow cytometry analysis showed that the proportions of cells in the S phase (35.96% ± 1.53%) and G2/M phase (25.53% ± 1.33%) in the *Salmonella*-infected group were much higher than those in the S phase (28.63% ± 0.52%) and G2/M phase (15.46% ± 0.02%) in the uninfected group, thus indicating that the infection could induce cells to undergo rapid cell proliferation (S6A Fig). However, no significant difference was observed in the cell cycle distribution in HCT116 cells stably expressing CDC42 WT, CDC42 K153Q or CDC42 K153R (S6B Fig). By performing the EdU assay, we found that the proliferation activity of the infected HCT116 cells was significantly stronger than that of the uninfected group (S6C Fig), while K153 acetylation had no significant effect on cell proliferation activity (S6D Fig).

We then tested whether *Salmonella* infection could mediate the migration/invasion ability of cancer cells and whether CDC42 K153 acetylation participates in regulating this process. Consistent with the results of a previous study [65], *Salmonella* infection significantly enhanced the migration and invasion abilities of HCT116 cells (Fig 6D). CDC42 K153R-expressing cells showed a higher invasion ability than CDC42 K153Q-expressing cells and a higher migration ability than CDC42 WT-expressing cells (Fig 6E).

Enhanced cell migration is usually associated with the abnormal expression of the matrix metalloproteinase (MMP) family of proteins and cadherin. For example, gelatinases MMP-2 and MMP-9 are highly expressed in human cancer cells [66], and E-cadherin is mainly expressed in epithelial tissues and has a low expression level in cancer cells [67]. We used quantitative real-time PCR (qPCR) to determine the expression level of *mmp-2*, *mmp-9* and *e-cadherin* in HCT116 cells during *Salmonella* infection and found that the expression level of *mmp-2* and *e-cadherin* decreased, while the expression level of *mmp-9* increased significantly (Fig 6F). Moreover, protein levels of MMP-2, MMP-9, and E-cadherin were determined by western blot assay. Compared to the uninfected group, the expression levels of MMP-2 and MMP-9 were increased, while the expression level of E-cadherin was reduced in the infected group (Fig 6G). We then performed qPCR to investigate the expression levels of *mmp-2*, *mmp-9*, and *e-cadherin* in HCT116 cells stably expressing CDC42 WT, CDC42 K153Q, or CDC42 K153R. The transcription levels of *mmp-2* and *mmp-9* were significantly increased and that of *e-cadherin* was significantly decreased in CDC42 K153R-expressing cells as compared to those in the CDC42 WT and CDC42 K153Q groups (Fig 6H). These results were further confirmed by western blot assay (Fig 6I). The abovementioned findings suggest that the acetylation of CDC42 K153 was a potential regulatory factor of colon cancer cell invasion.

## Regulation of K153 acetylation in CRC

*Salmonella*-infected mice have been used as a model to study the relationship between bacterial infection and CRC development [68]. We then used a *Salmonella* infection mouse model to investigate the effect of CDC42 K153 acetylation on CRC tumorigenesis. Mice were infected with *Salmonella* by oral gavage followed by AOM-DSS treatment to induce CRC. The protocol for establishing the mouse model is summarized in Fig 7A. Colorectal tissue samples were collected on week 14, and the presence of *Salmonella* in the intestine was confirmed by qPCR (Fig 7B). Representative colons with tumors are shown in Fig 7C. Tumors in the AOM/DSS-treated mice group with *Salmonella* infection were located more proximally in the colon than those in the uninfected mice group. This result was consistent with the finding of a previous study, which demonstrated that *Salmonella* infection enhances tumorigenesis and tumor burden in the AOM/DSS mouse model [68]. Immunofluorescence staining with anti-*Salmonella*

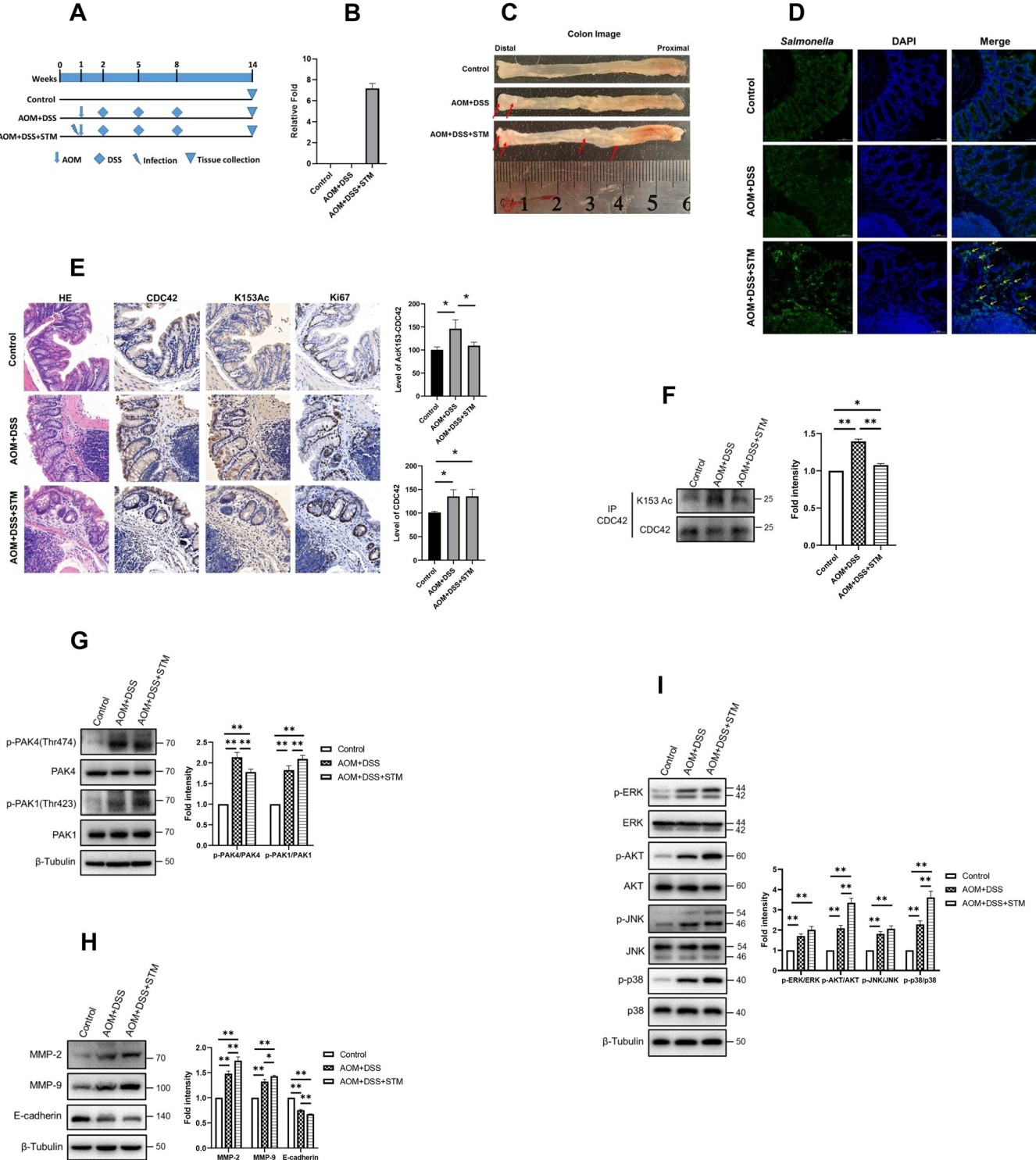

**Fig 7. K153 acetylation is down-regulated in CRCs.** (A) Scheme of the experimental design. Arrow symbols indicate azoxymethane (AOM), 10mg/kg body weight, administered through intraperitoneal injection. The lightning symbol indicates that mice were infected with *Salmonella* through a blunt gavage needle. Rhombus symbols indicate the administration of 1.5% dextran sodium sulfate (DSS) in drinking water. Triangle symbols indicate that mice were sacrificed, and their colorectal tissues were collected. (B) Fecal *Salmonella* was detected by PCR with *Salmonella*-specific primers after 14 weeks post-infection. (C) Colonic tumors *in situ*. Representative colons from the indicated groups 14 weeks after *Salmonella* infection. Tumors are indicated by red arrows. (D) Localization of *Salmonella* in the tissues of the indicated groups was assessed by immunofluorescence staining with anti-*Salmonella*-specific LPS antibodies. (E) Control mucosa and colonic tumors in each group were stained by hematoxylin-eosin (HE) staining, and the expression levels of CDC42, CDC42 K153 acetylated and

Ki67 in the indicated groups were determined by immunohistochemistry (IHC) and quantitated by modified H-score (left panel). Quantitative imaging of CDC42 K153 acetylation (upper right panel) and CDC42 (lower right panel) in untreated (control) mucosa and colonic tumors. Data are expressed as mean ± SD of control mice (n = 3) or AOM+DSS mice (n = 3) or AOM+DSS+STM mice (n = 4) in each group. Mouse tissues were harvested after 14 weeks post-infection, and levels of CDC42 K153 acetylation (F), p-PAK4, PAK4, p-PAK1 and PAK1 (G), MMP-2, MMP-9 and E-cadherin (H) were analyzed by WB (Left), and densitometry analysis was shown (Right). (I) p-ERK, ERK, p-AKT, AKT, p-JNK, and JNK in control and AOM/DSS induced tumors were analyzed by WB (Left), and densitometry analysis was shown (Right). $^*p<0.05$, $^{**}p<0.01$.

lipopolysaccharide (LPS) antibody revealed that bacteria were present in tumors from *Salmonella*-infected mice (Fig 7D). Tumors of the infected groups were confirmed by hematoxylin-eosin (HE) staining (Fig 7E). Immunohistochemistry (IHC) showed that the acetylation level of CDC42 K153 in tumors with *Salmonella* infection was lower than that in tumors without infection (Fig 7E). To further confirm the acetylation level of CDC42 K153 in mouse colon tissues, we extracted total protein from colon tissues and performed IP with anti-CDC42 antibody, and K153 acetylation level was determined by western blot assay. The results showed that K153 acetylation level was lower in the AOM/DSS-treated mice group with *Salmonella* infection than in the uninfected mice group (Fig 7F).

We also measured the activity of PAK1 and PAK4 by using their specific phosphorylation antibodies in all groups. The results of western blot assay showed that PAK4 phosphorylation level appeared to be lower in the AOM/DSS-treated mice group with *Salmonella* infection than in the uninfected mice group, whereas PAK1 phosphorylation level showed the opposite trend (Fig 7G). This finding is consistent with our aforementioned observation in the cell culture model that lower acetylation level of CDC42 K153 parallels with reduced PAK4 activity (Fig 5D).

To identify the effect of *Salmonella* infection on tumor development, western blot assay was performed to detect the expression of the MAPK/AKT signaling pathway. *Salmonella* infection increased the phosphorylation levels of ERK, AKT, JNK and p38 in tumors (Fig 7I). Moreover, high expression of MMP-2 and MMP-9 was observed in *Salmonella*-infected tumor tissues, while E-cadherin expression was attenuated in tumor tissues (Fig 7H). These findings suggest that the manifold effect of *Salmonella* infection on tumor development in mouse model, and CDC42 K153 acetylation may play an important role in this process.

To determine CDC42 K153 acetylation level in human CRCs, we collected paired CRC tissues and adjacent normal tissues from 17 patients with CRC. CDC42 expression level and CDC42 K153 acetylation level were determined by IHC, and a representative image of IHC was shown in Fig 8A. Compared to adjacent normal tissues, higher expression of CDC42 was observed in 64.7% (11/17) of CRC tissues, while lower K153 acetylation level was noted in 88.2% (15/17) of CRC tissues (Fig 8B). The presence of bacteria in tissues was assessed by IHC with *Salmonella*-specific LPS antibody. Bacterial cells were present in 60% (9/15) of CRC tissue samples with low K153 acetylation level.

To further confirm whether CDC42 K153 maintains a low level of acetylation in human CRC tissues, we collected 69 pairs of CRC tissues and adjacent normal tissues from CRC patients presented with different tumor stages, including 18 tissue pairs of patients in stage I, 19 tissue pairs of patients in stage II, 26 pairs of patients in stage III, and 6 pairs of patients in stage IV. The survival rates of these patients are shown in S7A Fig. IHC showed that K153 acetylation level in CRC tissues was significantly lower than that in adjacent normal tissues ($p = 0.0001$) (S7B Fig), and CDC42 K153 acetylation levels were strongly correlated with CRC stage. For samples in stage I, K153 acetylation levels in CRC tissues were slightly weaker than those in adjacent normal tissues ($p = 0.1681$), while for samples in stage II ($p = 0.0074$), III ($p = 0.0154$), and IV ($p = 0.0081$), K153 acetylation levels in the malignant tissues were significantly lower than those in adjacent normal tissues (Figs 8C and S7C). We also found that

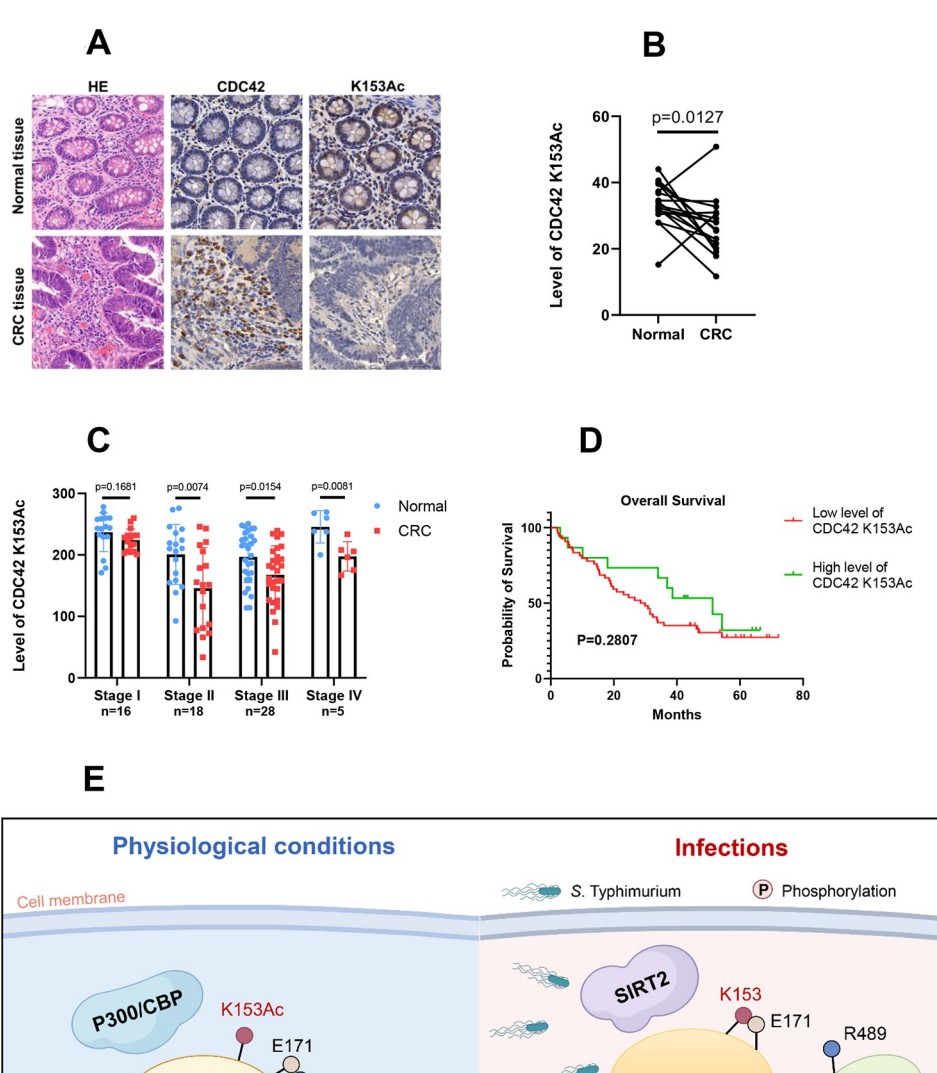

**Fig 8. Low level of CDC42 K153 acetylation predicts poor overall survival of patients with CRC.** (A) Human CRC specimens were confirmed by HE staining, and the expression levels of CDC42 and CDC42 K153 acetylation and the localization of bacteria in tissues of 17 human CRC specimens were detected by IHC. CDC42 K153 acetylation level was significantly lower in the colorectal adenocarcinoma tissues than in the adjacent normal colorectal tissues as determined by IHC. The average of IHC intensity ± SD were quantitated by modified H-score from two groups of 17 patient samples (B) and 69 patients with different CRC stages (C) is shown. (D) Kaplan–Meier analysis of overall survival of 69 patients with CRC according to CDC42 K153 acetylation level. (E) Model of CDC42 K153 acetylation effect on colorectal tumorigenesis. Under physiological conditions, acetylated CDC42 K153 can maintain the normal physiological function of cells through MAPK phosphorylation (including ERK, JNK and p38) mediated by the CDC42-PAK4 signaling pathway. When *Salmonella* infects a host cell, SIRT2 is activated to deacetylate CDC42 K153, which causes an impaired binding of its downstream effector PAK4 and an attenuated phosphorylation of p38 and JNK, consequently reduces cell apoptosis. Moreover, low acetylation level of CDC42 K153 may contribute to the migration and invasion abilities of CRC cells, and promoting tumorigenesis, which may activate tumors mainly through the CDC42-PAK signaling axis.

patients with lower level of K153 acetylation appeared to have worse survival rates, though not reaching statistical significance (Fig 8D). These results suggest that CDC42 K153 acetylation is associated with the poor prognosis of patients with CRC.

## Discussion

In the present study, acetylation was identified as a novel regulator of CDC42 during bacterial infection. On the basis of this finding, we propose a model wherein acetylated CDC42 K153 mainly maintains the normal functions of cells through the CDC42-PAK4 signaling pathway under physiological conditions. *Salmonella* infection activates SIRT2 to deacetylate CDC42 K153, reduces cell apoptosis, enhances the migration and invasion abilities of tumor cells, and promotes tumorigenesis (Fig 8E).

Recently, it has been increasingly recognized that tumors contain various commensal microorganisms, and these bacteria are mainly opportunistic and pathogenic species [5]. Our present study showed that several pathogenic bacteria could reduce the acetylation of CDC42 K153, while some probiotics did not change CDC42 K153 acetylation level. *Salmonella* could utilize the Type 3 Secretion Systems (T3SSs) to manipulate host proteins to facilitate infection of host cells. However, *F. nucleatum*, *E. faecalis* and *L. monocytogenes* do not possess T3SSs. Although EPEC harbors T3SS, it does not share same effectors with *Salmonella*. Thus, it is likely that the reduction of K153 acetylation level caused by bacterial infection does not depend on the T3SSs, but other unknown factors.

Previous studies showed that bacteria interfered with the fate of host cells by hijacking the Rho GTPases, thereby promoting the survival or death of host cells based on their needs [16]. Enteric pathogens such as *Shigella*, *Salmonella*, and EPEC share convergent targeting strategies to hijack host responses, and the actin network is one of their main targets [16]. The T3SSs of enteric pathogens can secrete various effector proteins to activate small GTPases to promote bacterial invasion [69]. For example, *Salmonella* infection can directly activate CDC42 through the effector protein SopE2 (*Salmonella* T3SS-1 effector) to promote actin cytoskeleton rearrangement, thereby facilitating its own invasion [70]. Bacterial infection can induce the expression of the host protein deacetylase SIRT2 to modulate the host immune responses by deacetylating substrate proteins [47,48]. In line with this finding, we observed that *Salmonella* infection could affect the binding of CDC42 and PAK4 by reducing the acetylation level of CDC42 K153 through SIRT2. Therefore, *Salmonella* can manipulate the activity of CDC42 through the following two completely different mechanisms during infection: one is the direct activation of CDC42 by secreting the effector protein SopE2; and the other one is the regulation of the CDC42-PAK signaling axis by decreasing the acetylation level of CDC42 K153. In this study, although the acetylation of CDC42 K153 is recognized as a key step regulating the interaction between CDC42 and PAK4, other *Salmonella* infection-induced PTMs of CDC42 may also change the affinity between these two proteins. Moreover, bacterial component(s) involved in the regulation of CDC42 K153 acetylation are still unknown. These issues are worth exploring in the future.

The intratumoral microorganisms can persist during metastasis and passage in CRC [71], and tumor bacteria can contribute to the metastasis and colonization of tumor cells [72]. A small number of intratumoral microorganisms can promote tumor metastasis by modulating cellular cytoskeleton and cell viability [72]. Malignant tumors have a strong ability to metastasize, but drastic tumor cell death occurs when tumor cells reach the distant organs [73]. One of the reasons is that apoptosis might be induced in those tumor cells [74]. Therefore, apoptosis serves as a natural barrier to cancer development, and resistance to apoptosis can help tumor cells

to survive after metastasis. Both JNK and p38 pathways are involved in the survival of tumor cells, and inhibition of p38 and JNK phosphorylation could reduce tumor cell apoptosis [58].

*Salmonella* can regulate cell apoptosis through the MAPK pathway. For example, either the sustained activation of AKT by SopB or the blockade of the JNK pathway by AvrA prevents cell apoptosis [75,76]. In consistent with this, we observed attenuated JNK and p38 activation in cells expressing non-acetylated CDC42 K153 mimics, suggesting that low acetylation level of CDC42 K153 could inhibit CRC cell apoptosis. However, under *Salmonella* infection condition, acetylation level of CDC42 K153 was correlated to the phosphorylation of p38 instead of JNK. Cells transfected with CDC42 K153Q or CDC42 K153R showed higher JNK phosphorylation levels compared to CDC42 WT transfected cells, which is contrary to the fact that K153R inhibited JNK phosphorylation in uninfected cells. One explanation is that as a complex condition, bacterial infection eliminates the phosphorylation change of JNK caused by acetylation of CDC42 K153. This phenomenon suggests that JNK signaling pathway is important during infection and may have more complex regulatory signals than p38 signaling pathway. Moreover, the mimicked acetylation and non-acetylation by amino acids substitution only simulate the state of acetylation to a certain extent, which may have limitations.

Activation of CDC42 facilitates tumor cell survivability and microadenoma formation, so CDC42 is considered to be crucial for tumorigenesis [14]. Moreover, CRC cells overexpressing CDC42 K153R showed a stronger migration and invasion ability, thus suggesting that the low acetylation level of CDC42 K153 may contribute to the migration and invasion of CRC cells. Therefore, both the expression level of CDC42 and the acetylation of CDC42 K153 play an important role in the development of CRC. CDC42-PAK1 is an important signaling pathway involved in tumor formation. In the AOM/DSS mouse model, lower CDC42 K153 acetylation caused by *Salmonella* infection is associated with higher PAK1 phosphorylation but lower PAK4 phosphorylation. Thus, the CDC42-PAK signaling axis played a crucial role in tumor formation during bacterial infection. Our results showed that the acetylation of CDC42 has little effect on the binding of PAK1, but the regulatory mechanism in the AOM/DSS induced CRC model with *Salmonella* infection is highly complicated, so other factors might enhance phosphorylation of PAK1 besides CDC42. Moreover, studies show that *Salmonella* infection could trigger the blooms of the resident commensal *Enterobacteriaceae* [77,78], which suggests that endogenous enterobacteria may jointly contribute to the development of CRC in the presence of pathogenic bacteria such as *Salmonella*.

CDC42 is highly expressed in 60% CRC, and its level is positively correlated with poorly differentiated CRCs [13]. Many amino acid residues of CDC42 are prone to mutation and are involved in tumorigenesis [79], so inhibition of CDC42 or components of its signaling pathway is considered an attractive cancer therapeutic target. Phosphorylation of CDC42 at tyrosine 64 can promote its interaction with the Rho-GDP dissociation inhibitor [80], while phosphorylation at serine 71 can reduce the cytopathic effect induced by *Clostridioides difficile* [81]. As a member of Ras-subfamily, the small GTPase CDC42 is acetylated at K153 in six human cell lines and 11 rat tissues, thus suggesting that lysine-acetylation is probably a regulator of the function of CDC42 [82]. In the present study, we showed that CDC42 K153 acetylation level was generally lower in human CRC tissues than in adjacent normal intestinal tissue. These results suggest that the low levels of K153 acetylation may contribute to tumorigenesis. Moreover, the low level of CDC42 K153 acetylation in CRC predicted a poor outcome for the patients, thereby suggesting that CDC42 K153 acetylation may have a prognostic value for patients with CRC. Thus, these results indicate a gain-of-function of CDC42 acetylation in human CRC progression.

In summary, we found that *Salmonella* infection decreased the acetylation level of CDC42 K153 through SIRT2, and the low acetylation level of CDC42 K153 could block the interaction

between CDC42 and PAK4. Hypoacetylated CDC42 K153 exerted a gain-of-function in cells, and thus promoted CRC progression. Our study showed that bacterial infection regulated host cell signaling by manipulating the acetylation of a key protein, and thus, it provided new insights into the mechanism by which bacteria affect tumor development by regulating non-histone PTMs. In the future, in patients with hypoacetylated colon cancer, the efficacy of reversing K153 acetylation level, probably by inhibiting SIRT2, can be tested as an adjuvant therapy for CRC.

## Materials and methods

### Ethics statement

All animal procedures were approved by Shanghai Jiao Tong University School of Medicine, and this study was carried out in strict accordance with the National Research Council Guide for Care and Use of Laboratory Animals [SYXK (Shanghai 2007–0025)]. All surgery was performed under sodium pentobarbital anesthesia, and all efforts were made to minimize suffering.

### Reagents and antibodies

Anti-CDC42 (#10155-1-AP, 1:1000 for WB), anti-PAK4 (#14685-1-AP, 1:1000 for WB), anti-PAK1(#21401-1-AP, 1:1000 for WB), monoclonal anti-GST(#66001-2-Ig, 1:10000 for WB), polyclonal anti-HA (#51064-2-AP, 1:5000 for WB), monoclonal anti-SIRT2 (#66410-1-IG, 1:10000 for WB), anti-beta tubulin (#10094-1-AP, 1:2000 for WB), MMP-2 (#10373-2-AP, 1:1000 for WB), MMP-9 (#10375-2-AP, 1:1000 for WB) were purchased from Proteintech; Monoclonal anti-HA (#AB0025, 1:5000 for WB); E-cadherin (#BS1098, 1:1000 for WB) were from Bioworld; Monoclonal anti-CDC42 (#sc-8401, 2 µg per 500 µg of total protein for Immunoprecipitation), anti-p-PAK4 (sc-135775, 1:200 for WB) were from Santa Cruz; Anti-p-PAK1 (#2601, 1:1000 for WB), anti-p-AKT (#4060, 1:1000 for WB), anti-AKT (#4691, 1:1000 for WB), anti-p-ERK (#9101, 1:1000 for WB), anti-ERK (#4370, 1:1000 for WB), anti-p-p38 (#9211, 1:1000 for WB), anti-p38 (#9212, 1:1000 for WB), anti-p-JNK (#9251, 1:1000 for WB), anti-ERK (#9252, 1:1000 for WB), anti-caspase 3 (#14220, 1:1000 for WB) and anti-PARP (#9542, 1:1000 for WB) were from Cell Signaling Technology; Monoclonal anti-HA (#M180, 1:500 for Immunoprecipitation), monoclonal anti-Flag (#M185, 1:500 for Immunoprecipitation), polyclonal anti-Flag (#PM020, 1:1000 for WB) and deacetylase inhibitors TSA (#9950) were from MBL; Deacetylase inhibitor NAM (#N1651) was from APExBIO; CBP/p300 inhibitor A-485 (#N1651) was from MCE; SIRT2 inhibitor AK7 (#4754–10) was from Tocris Bioscience; Staurosporine (#abs810006) was from Absin; Polybrene (#H9268) and puromycin (#P8833) were from Sigma-Aldrich. The anti-CDC42 K153Ac specific polyclonal antibody was raised against a synthetic peptide (Jiaxuan Biotech). In brief, the immune peptide DLKAVK(Ac)YVECSALTQKGL was used as antigen to immunize rabbits. During 45 days, rabbits were immunized for four times, and the antiserum was collected, and control peptide DLKAVKYVECSALTQKGL was used to remove non-specific antibody. The sensitivity and specificity of antibody were evaluated by ELISA and WB.

### SILAC labeling and Mass spectrometry analysis

For SILAC experiments, THP-1 or U937 cells were cultured in RPMI 1640 deficient in L-arginine and L-lysine, and supplemented with 10% dialyzed FBS. Two linages of cells were cultured in light medium (L-arginine (Arg 0) and L-lysine (Lys 0)) and heavy medium (L-arginine 13C6-15N4-HCl (Arg 10) (#89990, Thermo) and L-lysine 13C6-15N2-HCl (Lys 8))

(#88209, Thermo). Prior to infection, cells were treated with PMA (Phorbol 12-myristate-13-acetate) (#P1585, Sigma-Aldrich) for 24 h, then heavy-labeled cells were infected at 90% confluency with *S.* Typhimurium 14028S at MOI 100, while light-labeled cells were mock infected. After incubation for 1 h at 37˚C in a 5% $CO_2$ atmosphere, extracellular bacteria were killed at 100 mg/ml gentamicin for 2 h, and then switched to medium containing 25 mg/ml of gentamicin for the remainder of the experiment. Cells were lysed at indicated time points in lysis buffer containing 8M urea, 50 mM Tris-HCl pH 8.0, 150 mM NaCl, 1 mM ethylene diamine tetra acetic acid (EDTA), 2 g/l Aprotinin, 10 g/l Leupeptin, 1×protease inhibitor cocktail (#CW2200, Cwbio), and 5 mM sodium butyrate. Protein concentration was determined using a BCA protein assay and 5 mg proteins in two states mixed at equal ratio. Samples were digested with trypsin, and peptides were enriched for lysine acetylation using the anti-KAc antibodies noncovalently coupled to protein A agarose beads (#13416, Cell Signaling Technology). Desalted peptides were analyzed by online nanoflow liquid chromatography tandem mass spectrometry (LC-MS/MS) using a Q Exactive mass spectrometer. Heavy arginine (Arg10) and lysine (Lys8) were selected for SILAC quantification. The eXtracted Ion Current (XIC) of all isotopic clusters associated with the identified amino acids sequence in the light label cluster was summed up, and calculated the ratio between two heavy and light label partners. Then the ratio was normalized, the median of the total ratio population was shifted to 1 and data was analyzed using the MaxQuant software version 1.3.0.5. We used Significance A to assess the significance of outlier ratios. Only peptides with average 1.5-fold change of acetylation in two states and significance A values less than 0.05 were considered up or down regulated. The score at lysine acetylation of proteins greater than 20 is considered to be reliable.

## Plasmids construction

Flag-CDC42 was created by cloning human CDC42 protein ORF (GenBank: NM_001039802.2) into a pCDH-Flag vector using *Eco*RI and *Not*I restriction sites. Point mutations of CDC42 were generated by using the KOD-plus-mutagenesis Kit (#SMK101, TOYOBO). GST-CDC42 was generated by cloning human CDC42 ORF (GenBank: NM_001039802.2) into a pGEX-4T-1 vector using *Bam*HI and *Not*I restriction sites. HA-PAK4 and HA-PAK1 were constructed by cloning human PAK4 protein ORF (GenBank: NM_001014831.3) or PAK1 protein ORF (GenBank: NM_001128620.2) into a pKH3-HA vector using *Hind*III and *Xba*I restriction sites. Myc-Gcn5, HA-PCAF, HA-Tip60, HA-ACAT1, HA-SIRT2 and HA-SIRT1 were gifts from Prof. Shi-Min Zhao (School of Life Sciences and Institutes of Biomedical Sciences, Fudan University, Shanghai, China). HA-CBP and HA-p300 were gifts from Prof. Jian-Xiu Yu (Shanghai Jiao Tong University School of Medicine, China). V5-HDAC3, V5-HDAC6, V5-HDAC10, shRNA-CDC42 and shRNA-SIRT2 were purchased from DNA library (Shanghai Jiao Tong University School of Medicine, China).

## Bacterial strains

*S.* Typhimurium strain 14028S and strain SL1344, *Fusobacterium nucleatum* strain ATCC 25586, *Enterococcus faecalis* strain ATCC 29212, Enteropathogenic *Escherichia coli* (EPEC) strain E2348/69 and *Listeria monocytogenes* strain 10403S were used in this study.

## Cell culture and bacterial infection

HEK293T and HEK293FT cells were maintained in Dulbecco's modified Eagle medium supplemented with 10% fetal bovine serum (FBS). HCT116 cells were cultured in McCOY's 5A medium supplemented with 10% FBS. All cell lines were purchased from the Shanghai Institute of Cell Biology, Chinese Academy of Sciences (Shanghai, China). For western blot,

HEK293T and HCT116 cells were stimulated with *S.* Typhimurium strain 14028S (MOI 100) or indicated strains for 1 h, and the cells were lysed for further analysis.

## Transfection, Immunoprecipitation and Co-Immunoprecipitation

All transfections were performed by using lipofectamine 2000 (#11668030, Invitrogen). HEK293T cells transfected with indicated plasmids were lysed in lysis buffer (#1861603, Thermo) with protease inhibitor cocktail (#C600386, Sango). For Co-Immunoprecipitation, 1000 μg of total lysed proteins and 20 μl protein A/G-agarose beads (#20423, Thermo) were incubated with indicated antibodies at 4˚C overnight, then washed with lysis buffer and followed by western blot analysis.

## Lentivirus construction and infection

To establish the stable knockdown cell lines, shRNA-CDC42 or shRNA-SIRT2 was co-transfected with psPAX2 and pMD2G into HEK293FT cells (The shRNA targets the 3'UTR region rather than the gene coding region of genes, and the sequences of shRNA were shown in the S2 Table). To establish the stable HCT116 cells expressing CDC42 WT, CDC42 K153Q, CDC42 K153R, the pCDH-flag vector containing indicated gene was co-transfected with psPAX2 and pMD2G into HEK293FT cells. Two days after transfection, the virus particles were collected and filtered through a 0.45 μm filter. Indicated cells were infected with 8 mg/ml polybrene and selected with puromycin following the manufacturer's instructions.

## *In vitro* GST-CDC42 acetylation or deacetylation

The pGEX-4T-1-CDC42 plasmid was transformed to *E. coli* BL21 competent cells with 0.5 mM IPTG inducing for 4 h at 37˚C. Bacterial pellets were lysed in bacterial lysis buffer (50 mM Tris-HCl pH 7.5, 0.5 M NaCl, 10% glycerin, protease inhibitor cocktail (Cwbio). The lysate was incubated with 20 μl GST-Sefinose Resin 4FF (Settled Resin) (#C600031, Sango) overnight at 4˚C for the binding of fusion protein GST-CDC42 to the beads. For GST-CDC42 acetylation or deacetylation assays *in vitro*, HA-CBP/p300, or HA-SIRT2 cell lysates were purified from HEK293T cells. The lysates were incubated with GST-CDC42 bounded beads at 4˚C for 6 h, then the beads washed for five times by using the same lysis buffer and followed by western blot analysis.

## CDC42-GTP activity assay

HEK293T stable CDC42 knockdown cells were seeded in 6 cm dishes and transfected with CDC42 WT, K153Q or K153R plasmid. CDC42 Activation Assay Kit (#11859, Cell Signaling Technology) was used to determine CDC42 activity according to the manufacturer's instructions. Since GTP-bound CDC42 is active and can bind to intended proteins, the kit utilizes the CDC42 binding to PAK1 to determine the activity of CDC42. The kit uses PAK1 as an intended protein. Therefore, the activity of CDC42 is determined by measuring the amount of CDC42 bound to the beads. Specifically, cells were lysed in ice-cold lysis buffer with protease inhibitor cocktail (Sango) and cleared with glutathione-agarose beads. Cell lysates were incubated with GTP 15 min at 30˚C, then the lysates were incubated with PAK1 PBD agarose beads 1 h at 4˚C, followed by washing and pelleting. The beads were resuspended in sample buffer, and GTP-CDC42 was detected using Flag specific monoclonal antibody.

## RNA isolation and quantitative RT-PCR

Total cellular RNA was extracted using RNA Isolation Kit (#RC101, Vazyme). The cDNA was synthesized using the HiScript III RT SuperMix kit (#R323, Vazyme). Primers used are listed in S2 Table. GAPDH gene was used as an internal control for quantitative gene expression analyses.

## Cell cycle analysis, Apoptosis analysis, EdU assay, invasion and migration assay

To test the impact of infection, HCT116 cells were stimulated with *S*. Typhimurium strain 14028S (MOI 100) for 0.5 h and extracellular bacteria were killed by a 1 h incubation in 100 μg/mL gentamicin. To test the role of K153 acetylation, the indicated plasmids were transfected into the HCT116 CDC42 knockdown cells, followed by 48 h of incubation.

Cells were stained by cell cycle staining kit (#CCS012, Multi sciences) for cell cycle analysis and stained by Annexin V Apoptosis Detection Kits (#88-8007-72, Invitrogen) for apoptosis analysis, according to manufacturer's instructions followed by flow cytometry analysis. Cells were analyzed on a CytoFLEX LX Flow Cytometer (Beckman Coulter). Data analysis of the distribution of the different phases of cell cycle was performed using the ModFit LT software (Verity Software House Inc), at least 15,000 live singlet cells were considered for each subpopulation. Percentage of Annexin V+ Propidium iodide (PI)- or Annexin V+PI+cells were analyzed by using CytExpert software, at least 10,000 cells were collected.

For Edu assay, all cells were seeded at a density of $3 \times 10^5$ cells per well into 24 well plates, and proliferation was determined by incorporation of 5-ethynyl-20-deoxyuridine (EdU) using an EdU cell proliferation assay kit (Sango) following the manufacturer's instructions.

For the migration assay, all cells were incubated in serum-free medium before harvested. Seeded the $3 \times 10^4$ *Salmonella* infected HCT116 cells into the upper chamber of the Transwell apparatus (#3464, Corning Costar) and cultured for 24 h, while the HCT116 CDC42 knockdown cells expressing indicated plasmids were seeded at a density of $1 \times 10^5$ cells/ chamber and cultured for 36 h. Chambers was filled and cultured in medium with 10% FBS, while 20% FBS was added in the lower chamber. The cells on the top surface of the insert were removed with a cotton swab, and fixed by 4% paraformaldehyde. Then, the cells stained by 0.1% crystal violet, and five randomly selected fields were counted under a microscope. Cells seeded on matrigel (500 ng/mL; BD Biosciences) precoated chambers for the invasion assay, while the other steps were identical to those of the migration assay.

## CRC mouse model, clinical samples and histological testing

Animal experiments were performed as previously described [68]. Briefly, 8 weeks old specific pathogen-free female C57BL/6J mice were purchased from Shanghai Lingchang Experimental Animal Co., Ltd. Water and food were withdrawn 4 h before the mice were infected with 50 colony-forming units of *S*. Typhimurium SL1344 (100 μl suspension in saline) by oral gavage. Then, the mice were treated with 10 mg/kg of AOM (Sigma-Aldrich) by intraperitoneal injection. After one week, mice were provided with drinking water containing 1.5% dextran sodium sulfate (DSS) (MP Biomedicals) for 7 days, followed by another 14 days with water, and the cycle was repeated 4 times. The sample size was n = 6 mice in the control group with no treatment and n = 15 in each experimental group. After the final cycle, intestinal tissue samples were harvested for western blot or freshly fixed in 4% paraformaldehyde for 24 h and embedded in paraffin followed by histological testing.

Blocks of human CRC tissue were obtained from the Division of Gastroenterology and Hepatology, Renji Hospital, Shanghai, China. Blocks were stained at the Wuhan Servicebio

Technology Co., Ltd. The clinicopathological characteristics of patients with CRC were shown in S3 Table.

HE and IHC staining was performed by Wuhan Servicebio Technology Co., Ltd, the slides were stained with anti-CDC42 (Servicebio, #GB11570, 1:150), anti-Ki67 (1:500), anti-K153Ac (Servicebio, #GB111141, 1:50) anti-bacterial LPS (HycultBiotech, #HM6011, 1:1000), and immunohistochemical expressions were quantitated by modified H-score [83]. According to H-score, if the acetylation level of CDC42 K153 in tumor tissues is lower than that in adjacent tissues, it is defined as a low acetylation. Otherwise, it is defined as a high acetylation. Sections of mouse colonic tissue and colonic tumors were stained with anti-*Salmonella* LPS (Novus biologicals, #NB600-1087, 1:200) followed by immunofluorescence assay [68,84].

## Statistical analysis

All data were collected from more than 3 independent experiments. Data were analyzed using GraphPad Prism 8.0 software (GraphPad Software). Statistically significant differences were examined using Two-tailed Student's *t* test to derive the significance of differences between two groups. $p < 0.05$ was considered to be significant.

## Supporting information

**S1 Fig. *Salmonella* infection alters host protein acetylation.** (A) Increased and decreased H: L plots for acetylated peptides in THP-1 and U937 cells are shown. (B) Characterization of CDC42 specific acetyl-K153 antibody. Specificity of CDC42 K153 acetyl-antibody was determined by dot blot assay. A nitrocellulose membrane was spotted with different amounts of acetyl-K153 peptide or unmodified peptide, and detected with purified K153 Ac antibody or unpurified antiserum, respectively. (C) K153 acetylation and several pathogenic bacteria and gut-friendly bacteria. HEK293T cells overexpressing CDC42 were infected by five pathogenic bacteria (*S.* Typhimurium, *F. nucleatum*, *Enterococcus faecalis*, Enteropathogenic *Escherichia coli* (EPEC) and *L. monocytogenes*) or gut-friendly bacteria (*Bifidobacterium adolescentis*, *S. thermophiles*, *L. rhamnosus*, *L. plantarum* and *L. paracasei*) at MOI of 100 individually, and the cells were harvested for further IP and WB experiments after 1 h. K153 acetylation was measured by IP with anti-Flag antibody and followed by WB with anti-K153Ac.
(TIF)

**S2 Fig. Acetylation of K153 is essential for CDC42 interaction with PAK4.** (A) HEK293T-shCDC42 and HEK293T-shRNA vector control (shRNA NC) stable cell lines were established. CDC42 was knocked down in HEK293T cells by using several shRNAs. The protein expression levels were detected by WB with anti-CDC42 antibodies. (B) Activation of CDC42 by *Salmonella* infection. GTP-bound form of Flag-tagged CDC42 was transfected in HEK293T cells followed by *Salmonella* infection, and the expression levels of GTP-CDC42 were compared using PAK1-PBD IP and WB.
(TIF)

**S3 Fig. CDC42 K153 is deacetylated by SIRT2.** HA-SIRT2 (A), HA-SIRT1 (B), V5-HDAC3 (C), V5-HDAC6 (D), or V5-HDAC10 (E) was individually co-transfected with Flag-CDC42 into HEK293T cells. Cell lysates were used for IP with anti-Flag antibody or normal mouse IgG (as a negative control) and then analyzed by WB with the indicated antibodies. (F) SIRT2 was knocked-down in HEK293T cells by using several shRNAs. The protein levels were detected by WB with anti-SIRT2 antibodies.
(TIF)

**S4 Fig. CDC42 K153 is acetylated by p300/CBP.** CDC42 K153 acetylation and PCAF, Tip60, ACAT1 and Gcn5. HA tagged PCAF (A), Tip60 (B), ACAT1 (C), or Myc-tagged Gcn5 was individually co-transfected with Flag-CDC42 into HEK293T cells. K153 acetylation of was determined by IP with anti-Flag antibody, while normal mouse IgG was used as a negative control, followed by WB with the indicated antibodies.
(TIF)

**S5 Fig. Acetylation of CDC42 K153 affects of JNK and p38 phosphorylation.** (A) HEK293T-shCDC42 and HEK293T-shRNA NC stable cell lines were established. CDC42 was knocked down by two shRNAs in HCT116 cells. (B) HEK293T-shPAK4 and HEK293T-shRNA NC stable cell lines were established. CDC42 in HCT116 cells was knocked-down by several shRNAs.
(TIF)

**S6 Fig. Unacetylated CDC42 K153 promotes the invasion ability of HCT116 cells.** (A) Percentage of HCT116 cells in the different stages of the cell cycle after stimulation with (STM) or without (UI) *S*. Typhimurium. Cell cycle stage distribution was determined by flow cytometry analysis after PI staining. Representative cell cycle profiles for 24 h.p.i. are shown. (B) Percentage of HCT116 cells stably expressing CDC42-WT, CDC42 K153Q, or CDC42 K153R in the different stages of the cell cycle. EdU proliferation assay of the effect on *S*. Typhimurium-stimulated HCT116 cells (C) and CDC42 WT or QR mutants of CDC42 K153 in HCT116 cells (D). $^{**}p<0.01$.
(TIF)

**S7 Fig. Low level of CDC42 K153 acetylation predicts poor overall survival of patients with CRC.** (A) Kaplan-Meier analysis of overall survival rates of CRC patients at different stages of CRC. (B) CDC42 K153 acetylation level was significantly lower in colorectal adenocarcinoma tissues than in adjacent normal colorectal tissues as determined by IHC. The average value of IHC intensity ± SD were quantitated by modified H-score from two groups of 69 patient samples is presented. (C) CDC42 K153 acetylation level of 69 human CRC specimens with different CRC stages was detected by IHC.
(TIFF)

**S1 Table. Changed acetylation of THP-1 and U937 proteins were detected by SILAC.**
(XLSX)

**S2 Table. Sequence of shRNA and primers.**
(DOCX)

**S3 Table. Clinical characteristics of the colorectal cancer (CRC) patients.**
(DOCX)

## Acknowledgments

We would like to thank Jian-Xiu Yu, Xian Zhao, Xiao-Yan Yu and Hai-Yan Cai from Shanghai Jiao Tong University School of Medicine for technical support.

## Author Contributions

**Conceptualization:** Dan-Ni Wang, Jie Lu, Yu-Feng Yao.

**Formal analysis:** Dan-Ni Wang, Jin-Jing Ni.

**Funding acquisition:** Dan-Ni Wang, Jin-Jing Ni, Jie Lu, Yu-Feng Yao.

**Investigation:** Dan-Ni Wang, Jin-Jing Ni, Jian-Hui Li, Fang-Jing Ni.

**Methodology:** Dan-Ni Wang, Jin-Jing Ni.

**Project administration:** Dan-Ni Wang, Yu-Feng Yao.

**Resources:** Ya-Qi Gao, Zhen-Zhen Zhang, Jing-Yuan Fang.

**Supervision:** Jie Lu, Yu-Feng Yao.

**Validation:** Dan-Ni Wang.

**Writing – original draft:** Dan-Ni Wang, Jin-Jing Ni.

**Writing – review & editing:** Dan-Ni Wang, Jie Lu, Yu-Feng Yao.

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
