## [Decision Letter · Decision Letter 0]

8 Dec 2022

Dear Dr. Yao,

Thank you very much for submitting your manuscript "Bacterial infection promotes tumorigenesis of colorectal cancer via regulating" for consideration at PLOS Pathogens. As with all papers reviewed by the journal, your manuscript was reviewed by members of the editorial board and by several independent reviewers. In light of the reviews (below this email), we would like to invite the resubmission of a significantly-revised version that takes into account the reviewers' comments.

We cannot make any decision about publication until we have seen the revised manuscript and your response to the reviewers' comments. Your revised manuscript is also likely to be sent to reviewers for further evaluation.

Sincerely,

Zhaoqing Luo

Academic Editor

PLOS Pathogens

Karla Satchell

Section Editor

PLOS Pathogens

Kasturi Haldar

Editor-in-Chief

PLOS Pathogens

orcid.org/0000-0001-5065-158X

Michael Malim

Editor-in-Chief

PLOS Pathogens

orcid.org/0000-0002-7699-2064

Reviewer's Responses to Questions

**Part I - Summary**

Reviewer #1: In the manuscript titled “Bacterial infection promotes tumorigenesis of colorectal cancer via regulating 2 CDC42 acetylation”, Wang et al. investigated how S. Tm infection altered the lysine acetylome and found that among 72 KAc sites, acetylation of K153 of CDC42 was reduced. The authors provided evidence showing that reduced acetylation levels of CDC42 K153 can be observed in HEK293T cells in response to diverse pathogens. They subsequently demonstrated that S. Tm infection led to reduced binding between CDC42 and PAK4 and suggested that this phenotype resulted from the decreased acetylation of CDC42 K153. Wang et al. went on and identified SIRT2 and p300/CBP as a deacetylase and acetyltransferases for CDC42, respectively. Further, the authors showed that CDC42 K153 acetylation levels correlate with the phosphorylation status of p38 and JNK, cell survival, tumor cell migration and invasion. Lastly, the authors explored the relationship between CDC42 K153 acetylation and tumorigenesis in the large intestine using a murine CRC model and patient data. Overall, the concept of this manuscript is interesting, and a significant proportion of the experimental evidence supports the authors' conclusions. However, the clinical relevance of the core concept, and the flaws in experimental design and interpretation have dampened the enthusiasm for this manuscript.

Reviewer #2: In this work Wang et al. characterize the role of de/acetylation of a Rho GTPase, cell division cycle 42 (CDC42), in regulating Salmonella-induced tumorigenesis. CDC42 is highly expressed in CRCs and has previously been identified as an important player in cancer development. The authors present data that the acetylation of CDC42 at lysine 153 was reduced during Salmonella infection of cancer cells, which may impair the binding of its downstream effector PAK4. CDC42 was deacetylated by SIRT2 and acetylated by p300/CBP. Non-acetylated CDC42 reduced apoptosis and promoted the migration and invasion ability of colon cancer cells by increasing the phosphorylation of p38 and JNK. In addition, they tested the correlation between bacterial-induced tumorigenesis and CDC42 K153 acetylation in mice model and in in patients with colorectal cancer. They propose a model in which Salmonella-mediated deacetylation of CDC42 regulates PAK pathway to promote the colorectal tumorigenesis.This work established a novel connection in bacterial infection, host PTM regulation, and clinical phenotypes.

**Part II – Major Issues: Key Experiments Required for Acceptance**

Reviewer #1: Major Issues:

1. While the overall concept is interesting, the clinical link between Salmonella and CRC is weak. Salmonella is rarely reported to be overrepresented in or contribute to CRC development.

2. The evidence provided by the authors is insufficient to support their conclusion that the reduced binding between CDC42 and PAK4 results from the weakened interaction of CDC42 E171 and PAK4 R489, which the authors claimed to be caused by the reduced acetylation of K153. The author noted that CDC42 is activated during S. Tm infection and showed that GTP-bound CDC42 WT and K153 mutants have a similar affinity towards PAK1. However, no such experiments with PAK4 were performed; therefore, it is unclear whether the activation or the reduced acetylation is responsible for reduced PAK4 binding. Additionally, it is unclear whether other PTMs cause CDC42 to display reduced affinity towards PAK4 during STM infection.

3. The relevance of the murine model presented in this paper is unclear. C57BL/6 mice are Nramp-/-, which makes them susceptible to S. Tm infection. When antibiotics such as streptomycin is used, they succumb to S. Tm infection in about a week. Suppose these mice are not pre-treated with antibiotics, as described in this paper. In that case, they are resistant to S. Tm colonization even with an inoculum 40-fold higher than described in this paper (REF: PMID: 12704158). It is, therefore, unclear whether S. Tm colonized the mice at all for the duration of the experiments, and if it did, how did the mice survive 14 weeks of infection? Moreover, the authors provided no evidence related to the S. Tm population in the gut at any time point of the experiments. Lastly, it is difficult to comprehend how S. Tm can be present inside these mice's tumors without systemic spread, which Nramp-/- C57BL/6 mice are highly susceptible to.

Reviewer #2: 1. In general, the introduction is not well organized, which makes the significance of the study unclear. Figure legends and methods are not clearly described. For example, how does GTP-CDC42 detect? How is CDC42 acetylation peptide calculated?

2. The differences between some western blot bands are hardly distinguished (eg. Figures 5D, 5F, 6I, 6J, 7F, 7G, 7I) while the authors declare there are significant decrease or increase. Attach histograms next to them will help the readers to understand the data more easily.

3. Due to the lack of proper positive controls in some assays, some sub-figures should be put together to claim the conclusions, such as Fig3B and FigS3A-3D, Fig4A-B and FigS4A-3D, Fig5B and FigS5B.

**Part III – Minor Issues: Editorial and Data Presentation Modifications**

Reviewer #1: Minor issues:

1. The authors did not cite the paper in which the structure of PAK4:CDC42 complex.

2. The authors suggested that reduced binding between CDC42 and PAK4 leads to reduced PAK4 activation and, therefore, the reduced phosphorylation of JNK and p38. However, this conclusion is not well supported because they provided no evidence showing PAK4 phosphorylation supports JNK and p38 phosphorylation.

3. The isotope-labeled experiment can benefit from a more explicit description in the main text.

4. The authors did not provide evidence why the shRNA knock down of endogenous CDC42 did not impact the expression of the mutants.

5. “CDC42 K153 116 acetylation may be universally present in nature” is a strong conclusion based on the alignment of sequences from five organisms.

Reviewer #2: 1. In Figure 1, the authors did not explain that why lysine-acetylome was conducted. Is the total acetylation pattern altered in host cells during Salmonella infection? This could be detected by immunoblotting analyses with specific anti-KAc antibody. Also, they did not explain clearly why selected CDC42 for further investigation among more than 50 candidates.

2. In Figure 1D, the shown MS/MS spectrums of modified peptides is incomplete. The coverage of amino acids is not enough. Besides, how acetylation modification rate is calculated? The authors could refer to related documents (PMID: 30323948, 32504010).

3. In Supplementary Figure 1B, the specificity of the antibody to K153-acetylated (K153Ac) CDC42 remains to be determined. To exclude the cross-link reaction with acetylation modification of the other sites within CDC42, it should be tested with a transfected K153 CDC42 mutant.

4. In Supplementary Figure 1C, the infection efficiency of different bacteria should be determined.

5. In Figure 2A，why CDC42 acetylation specifically affect CDC42 binding with PAK4 other than PAK1. Is there a conserved site in PAK1 and PAK4 in the protein structure?

6. In Figure 2C，it seems that S. Typhimurium infection also decreased the binding between CDC42 and PAK1 (lane 1 and lane 3).

7. SIRT2 deacetylates CDC42 K153. However, in Figure 3G, the CDC42 acetylation ration did not increase after knock-down of SIRT2 (Lane 2-3), which is contradictive with the conclusion of Figure 2E.

8. Non-acetylated CDC42 K153 suppresses phosphorylation of p38 and JNK (Figure 5B). However, acetylation of CDC42 K153 only regulated the phosphorylation of p38 but not JNK during Salmonella infection (Figure 5C). Instead, the phosphorylation level of JNK seems decreased in WT-CDC42 transfected cells than in CDC42 K153Q and in K153R group (lane3 in Figure 5C). How to explain this?

9. Figure 5G-5I only indicated that non-acetylated CDC42 K153 inhibited the STS-induced apoptosis. However, there is a gap to say “Our results suggested that Salmonella could also prevent CRC cell apoptosis by decreasing CDC42 acetylation.” (Line 396).

10. Compared with WT, the expression of mmp-2 and mmp-9 increased both in K153Q and K153R group. The author should explain this.

11. PAK1 phosphorylation level appeared to be higher in the AOM/DSS-treated mice group with Salmonella infection. However, CDC42 acetylation does not affect its binding with PAK1 (Figure 2D), nor control the phosphorylation of PAK1 (Figure 5D). The author should explain this.

12. Line 327-329, expression and of CDC42 increased of CRC tissues. In the meanwhile, K153 acetylation level decreased. Author should discuss which factor determines the tumorigenesis.

13. In FigS6B, it is not clear how to define the high and low CDC42 K153 acetylation levels.

14. There is no obvious evidence in the figures to match the authors’ descriptions, such as line 330-332 and line 368-371.

15. In the discussion section, the author should also discuss the mechanism by which Salmonella induces the acetylation of CDC42 protein, such as bacterial derived LPS stimulation, iNOS pathway activation, or the direct effects of bacterial deacetylases like cobB. On the other hand, it should be considered that the effects of CDC42 K153 acetylation on Salmonella intracellular invasion and replication.

16. Line 150, CDC42 instead of CDC142; Line 281, K153 instead of K153R.

PLOS authors have the option to publish the peer review history of their article (what does this mean?). If published, this will include your full peer review and any attached files.

Reviewer #1: No

Reviewer #2: **Yes: **Shan Li
---

## [Decision Letter · Decision Letter 1]

3 Feb 2023

Dear Dr. Yao,

Thank you very much for submitting your manuscript "Bacterial infection promotes tumorigenesis of colorectal cancer via regulating CDC42 acetylation" for consideration at PLOS Pathogens. As with all papers reviewed by the journal, your manuscript was reviewed by members of the editorial board and by several independent reviewers. The reviewers appreciated the attention to an important topic. Based on the reviews, we are likely to accept this manuscript for publication, providing that you modify the manuscript according to the review recommendations.

Sincerely,

Zhaoqing Luo

Academic Editor

PLOS Pathogens

Karla Satchell

Section Editor

PLOS Pathogens

Kasturi Haldar

Editor-in-Chief

PLOS Pathogens

orcid.org/0000-0001-5065-158X

Michael Malim

Editor-in-Chief

PLOS Pathogens

orcid.org/0000-0002-7699-2064

Reviewer Comments (if any, and for reference):

Reviewer's Responses to Questions

**Part I - Summary**

Reviewer #1: (No Response)

Reviewer #2: The authors have answered all my questions. I have no more questions.

**Part II – Major Issues: Key Experiments Required for Acceptance**

Reviewer #1: (No Response)

Reviewer #2: (No Response)

**Part III – Minor Issues: Editorial and Data Presentation Modifications**

Reviewer #1: Minor issues:

1. While the authors provided some evidence supporting the roles of Salmonella in CRC, the overall link is still weak as compared to some other bacterial pathogens, such as Fusobacterium, which the authors noted. It will be helpful to include the information provided in the rebuttal letter in the Introduction section to better support the rationale underlying using Salmonella as the model pathogen.

2. The authors’ reasoning for using PAK1 as a control for their claims that “reduced acetylation of CDC42 weakens its interactions with PAK4”, is appreciated. However, the authors cannot exclude the possibility that other PTMs cause CDC42 to display reduced affinity towards PAK4 during STM infection. If the authors cannot provide additional evidence, the conclusion should be toned down.

3. The relevance of the mouse model still warrants questions. While the authors titrated the inoculum to optimize Salmonella infection, it is curious how S. Tm can infect B6 without antibiotic treatment, especially given the low inoculum dose (50 CFU). If the infection did lead to changes in tumorigenesis (either by incidence or pathology), logically, the infection should have led to some degree of inflammation, which was not reported in this study. In addition, it is unclear whether the mice used in this study contain Enterobacteriaceae (E. coli, Klebsiella, etc; if the mice were sourced from Jackson, they should be Enterobacteriaceae-free, but should have Enterobacteriaceae if they were sourced from Charles River or other vendors). Because the differences in 16S sequences between Salmonella and other Enterobacteriaceae members are so slight, it is unlikely to distinguish one from another using 16S qPCR. As such, if Salmonella infection leads to the bloom of endogenous Enterobacteriaceae (https://www.pnas.org/doi/10.1073/pnas.1113246109, https://www.cell.com/cell-host-microbe/pdfExtended/S1931-3128(18)30630-9), the detection method presented in this paper cannot distinguish what was detected. In summary, in the absence of additional supporting information, the relevance of this model should be toned down.

Reviewer #2: (No Response)

PLOS authors have the option to publish the peer review history of their article (what does this mean?). If published, this will include your full peer review and any attached files.

Reviewer #1: No

Reviewer #2: **Yes: **Shan Li

Figure Files:

Data Requirements:

Reproducibility:

References:

---

## [Editor Report · Decision Letter 2]

7 Feb 2023

Dear Dr. Yao,

We are pleased to inform you that your manuscript 'Bacterial infection promotes tumorigenesis of colorectal cancer via regulating CDC42 acetylation' has been provisionally accepted for publication in PLOS Pathogens.

Best regards,

Zhao-Qing Luo

Academic Editor

PLOS Pathogens

Karla Satchell

Section Editor

PLOS Pathogens

Kasturi Haldar

Editor-in-Chief

PLOS Pathogens

orcid.org/0000-0001-5065-158X

Michael Malim

Editor-in-Chief

PLOS Pathogens

orcid.org/0000-0002-7699-2064
---

## [Editor Report · Acceptance letter]

15 Feb 2023

Dear Dr Yao,

We are delighted to inform you that your manuscript, "Bacterial infection promotes tumorigenesis of colorectal cancer via regulating CDC42 acetylation," has been formally accepted for publication in PLOS Pathogens.

Best regards,

Kasturi Haldar

Editor-in-Chief

PLOS Pathogens

orcid.org/0000-0001-5065-158X

Michael Malim

Editor-in-Chief

PLOS Pathogens

orcid.org/0000-0002-7699-2064